# Atypical hippocampal excitatory neurons express and govern object memory

Adrienne I. Kinman[1], Derek N. Merryweather[1,7], Sarah R. Erwin[1,7], Regan E. Campbell[1], Kaitlin E. Sullivan[1], Larissa Kraus[1], Margarita Kapustina[1], Brianna N. Bristow [1], Mingjia Y. Zhang[1], Madeline W. Elder[1], Sydney C. Wood[1], Ali Tarik[1], Esther Kim[1], Joshua Tindall[1], William Daniels [1], Mehwish Anwer[2,3], Caiying Guo[4] & Mark S. Cembrowski [1,3,4,5,6] ✉

Classically, pyramidal cells of the hippocampus are viewed as flexibly representing spatial and non-spatial information. Recent work has illustrated distinct types of hippocampal excitatory neurons, suggesting that hippocampal representations and functions may be constrained and interpreted by these underlying cell-type identities. In mice, here we reveal a non-pyramidal excitatory neuron type − the "ovoid" neuron − that is spatially adjacent to subiculum pyramidal cells but differs in gene expression, electrophysiology, morphology, and connectivity. Functionally, novel object encounters drive sustained ovoid neuron activity, whereas familiar objects fail to drive activity even months after single-trial learning. Silencing ovoid neurons prevents non-spatial object learning but leaves spatial learning intact, and activating ovoid neurons toggles novel-object seeking to familiar-object seeking. Such function is doubly dissociable from pyramidal neurons, wherein manipulation of pyramidal cells affects spatial assays but not non-spatial learning. Ovoid neurons of the subiculum thus illustrate selective cell-type-specific control of non-spatial memory and behavioral preference.

The hippocampus, a brain region critical for spatial navigation and memory, has been richly studied for understanding how neuronal activity can represent features of the external and internal world. Historically, such representations have been examined through activity correlates of geographical space[1], with more recent evidence illustrating neural activity can also encompass a variety of other physical[2], temporal[3], affective[4,5], and abstract features[5,6] both in isolation and in conjunction. In collection, this evidence demonstrates that hippocampal activity can flexibly represent elements of the allocentric environment, as well as relational and subjective import of these elements. Whether such representations reflect a spatially tuned hippocampal network that is supplemented with non-spatial information, or embody separable spatial and non-spatial information streams, is a focal point of hippocampal research[7–10].

Pyramidal neurons of the hippocampus are frequently studied as the cellular substrate of hippocampal representations[1–6]. In the subiculum, the primary output of the hippocampus[11], pyramidal neurons have been shown to embody a variety of spatial and non-spatial receptive field properties. Individual cells of the subiculum can encode aspects of an animal's speed and trajectory[12–14], specific locations and elements of the spatial environment[12,15–18], context and choice[19], and a variety of aspects of memory[20–24]. As the subiculum relays such signals to a variety of downstream regions, this brain region offers a critical opportunity to interpret the content of signals

[1]Dept. of Cellular and Physiological Sciences, Life Sciences Institute, University of British Columbia, Vancouver V6T 1Z3, Canada. [2]Dept. of Pathology and Laboratory Medicine, University of British Columbia, Vancouver V6T 1Z7, Canada. [3]Djavad Mowafaghian Centre for Brain Health, University of British Columbia, Vancouver V6T 1Z3, Canada. [4]Janelia Research Campus, Howard Hughes Medical Institute, Ashburn 20147, USA. [5]School of Biomedical Engineering, University of British Columbia, Vancouver V6T 1Z3, Canada. [6]Department of Mathematics, University of British Columbia, Vancouver V6T 1Z2, Canada. [7]These authors contributed equally: Derek N. Merryweather, Sarah R. Erwin. ✉e-mail: mark.cembrowski@ubc.ca

that are relayed to extrahippocampal targets to drive hippocampal-dependent behavior.

At face value, the wide variety of receptive field properties of the subiculum may suggest that individual cells in the subiculum may flexibly represent multiple distinct types of information. However, recent experimental evidence has emerged that illustrates transcriptomically discrete subtypes of pyramidal neurons[25–28], which may constrain such flexibility and functional contributions[12,24,29]. Such potential subtype-specific functional specialization is also in agreement with recent results from artificial neural networks, which demonstrates that different neural architectures are optimal for disparate biologically-inspired computations[30]. As a consequence, key questions lie at the interface of the "bottom-up" cell-type heterogeneity of the subiculum and "top-down" perspectives of hippocampal representations. In particular, do discrete excitatory neuron subtypes have specialized feature selectivity and representational properties? If so, as neuron subtypes are typically amenable to specific access and manipulation, can these representations be linked to dissociable and causal contributions to behavior?

Here, using a cell-type-specific approach to merge "bottom-up" and "top-down" perspectives, we identify and interpret an anomalous excitatory neuron subtype in the subiculum. This non-pyramidal subtype, which exhibits an ovoid cell body and a variety of unique molecular, cellular, and circuit specializations relative to classical pyramidal cells, robustly responds to novel objects and fails to respond to familiar objects experienced in both the recent and remote past. Silencing ovoid neurons produces non-spatial object encoding deficits, and activating ovoid neurons produces behavioral seeking of previously experienced objects in both the recent and remote past. This work demonstrates that ovoid neurons can bidirectionally control memory-driven behavior across multiple behavioral timescales, in a dissociable manner relative to pyramidal neurons of the subiculum. As such, ovoid neurons extend the computational capabilities, timescales, and functions of the hippocampus.

## Results

### Identification and spatial registration of an outlying subiculum neuron subtype

As the subiculum encompasses a rich collection of transcriptomically defined excitatory neuron subtypes[25,26], we used a data-driven approach to identify any particularly outlying subiculum subtypes. Using published single-cell RNA sequencing data of putatively excitatory subiculum neurons[25], analysis of 1949 single-cell transcriptomes revealed one sparse and prominently outlying subtype relative to other subiculum neurons (UMAP dimensionality reduction and graph-based clustering at a coarse resolution: Fig. 1A). Such outlying cells expressed the neuronal marker *Snap25*, the excitatory neuron markers *Slc17a7, Slc17a6*, and *Camk2a*, and lacked expression of the inhibitory neuron marker *Gad1* (Fig. 1B, top; Supplementary Fig. 1A; Supplementary Fig. 2C–F). Expression of individual genes was sufficient to differentiate the outlying subtype from other subiculum neurons (e.g., enrichment of *Ly6g6e* and depletion of *Cck*; Fig. 1B, bottom; see also ref. 26), and a large number of genes were differentially expressed in the outlying subtype (n = 508 genes were differentially expressed at $p_{ADJ} < 0.05$, Fig. 1C). The outlying subtype was associated with a wide variety of neuronally relevant functions obtained from both Gene Ontology and KEGG Pathway analyses (e.g., chemical synaptic transmission, acetylcholine receptor signaling, oxytocin signaling, and calcium signaling: Supplementary Fig. 1B, C), suggesting a large degree of functional specialization. Analysis of a substantially larger scRNA-seq dataset of excitatory neurons, inhibitory neurons, and non-neuronal cells across the hippocampal formation and retrosplenial cortex (n = 11,812 cells total, from ref. 1) further confirmed that *Ly6g6e* expression was effectively restricted to excitatory neurons of the subiculum (Supplementary Fig. 2A–I).

As *Ly6g6e* exhibited strong and selective expression in the outlying subtype in this scRNA-seq dataset (>60-fold enrichment, $p_{ADJ} < 1e-78$: Fig. 1B), we next sought to validate this putative selectivity with other methods. To do so, we first confirmed and spatially mapped its expression relative to *Cck*. Using single-gene chromogenic ISH[31], *Cck* was found to be broadly expressed across the pyramidal cell layer, whereas *Ly6g6e* expression was associated with the deepest part of the subiculum (Fig. 1D–G; Supplementary Fig. 2J, K) (see also refs. 25,26). Fluorescent in situ hybridization (FISH), used to simultaneously detect expression of all these genes, confirmed that *Ly6g6e*-expressing cells also expressed the excitatory neuron marker *Slc17a7* and that *Ly6g6e* expression was mutually exclusive of expression of the inhibitory neuron marker *Gad1* (Fig. 1H, cf. Fig. 1B; see also Supplementary Fig. 3A–C). Moreover, *Ly6g6e* and *Cck* expression defined distinct cell types with abutting and largely non-overlapping spatial domains within the subiculum (Fig. 1I, J; see "Methods"), with *Ly6g6e*-expressing neurons located in the deepest part of the subiculum.

### Deep neurons are characterized by atypical non-pyramidal cell bodies

Several reports have illustrated that the deep subiculum is characterized by a polymorphic layer, distinguished by oval-shaped cell bodies oriented tangentially relative to pyramidal cells[32,33]. As the phenotypic identity of these atypical cells has not been identified, we next examined whether *Ly6g6e*-expressing cell bodies corresponded to non-pyramidal polymorphic cells. To do this, we performed FISH to identify subtypes (as in Fig. 1H–J), and used Nissl staining *post hoc* to label somata (Fig. 1K), which revealed that *Ly6g6e*-expressing neurons seemed to be smaller and tangentially oriented.

To analyze cell-body geometry and relate this geometry to gene expression, we first used DAPI and Nissl stains to identify nuclei and associated cell bodies, and manually segmented cell bodies by tracing individual cell bodies. From this, we examined cell-body properties relative to *Ly6g6e* and *Cck* expression (Fig. 1L). This analysis revealed that *Ly6g6e*-expressing cell bodies had significantly reduced overall area (Fig. 1M), along with other geometric differences (Supplementary Fig. 3D). To assess cell-body orientation, we measured aspect ratios registered to the alveus (Fig. 1N), and identified that *Ly6g6e*-expressing cell bodies were oriented tangentially and elongated along the alveus axis relative to *Cck*-expressing neurons (Fig. 1O). Performing an unbiased principal component analysis based upon multiple features measured from cell bodies (see "Methods"), we found that *Ly6g6e*-expressing cell bodies and *Cck*-expressing cell bodies could be generally separated based on their somatic geometry (Fig. 1P; data-driven cluster analysis: Supplementary Fig. 3B–G). To emphasize the differences between these atypical oval-shaped deep cells from classical pyramidal neurons, we hereafter refer to *Ly6g6e*-expressing cells as non-pyramidal "ovoid neurons".

### Ovoid neurons selectively target the anterior thalamus

To selectively access ovoid neurons, we generated a transgenic Ly6g6e-IRES-Cre mouse line (Fig. 2A; see "Methods"). Using Cre-dependent viral tracing to compare to relatively well-studied nucleus accumbens-projecting pyramidal neurons[2], we confirmed that Cre-expressing neurons occupied the deepest laminae of the subiculum (Fig. 2B, Supplementary Fig. 2L), and that *Cre* expression colocalized with *Ly6g6e* expression in Ly6g6e-IRES-Cre mice (Supplementary Fig. 3I–K). AAV-based viral reporting of Cre was used due to broad non-specific labeling in transgenic reporter crosses (Supplementary Fig. 4), likely due to transient expression of *Ly6g6e* in other cell types prior to maturity.

Following the long-range projections of ovoid neurons, we found such cells appeared to project solely to the anteroventral and anteromedial nuclei of the anterior thalamic nuclei (ATN; Fig. 2C, D and Supplementary Movie 1) (in agreement with refs. 34–36), and verified via retrograde tracing combined with mFISH that *Ly6g6e*-expressing

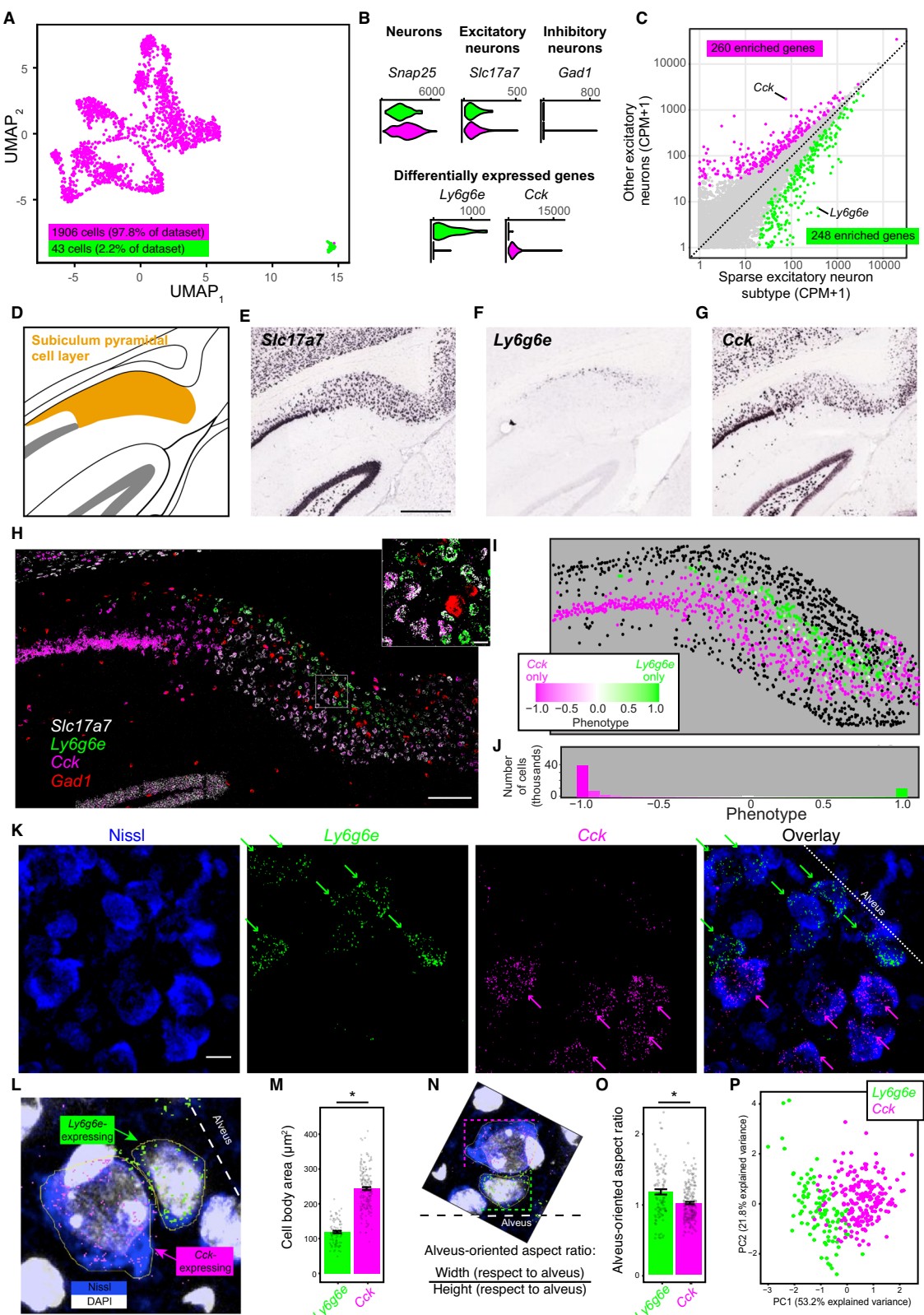

deep subiculum neurons formed ATN projections in the wild-type brain (Supplementary Fig. 3H). To validate and interpret these projections at a single-cell level, we next analyzed whole-brain axonal projections of individual neurons (via http://mouselight.janelia.org: ref. 37). From this dataset, we were able to identify subiculum neurons that projected to the ATN (e.g., Fig. 2E), as well as neurons that projected elsewhere (e.g., Fig. 2F). To analyze all projections in a systematic fashion, for each

subiculum projection neuron, we parcellated the brain into discrete regions and then calculated the length of axon in each region (see "Methods"). Dimensionality reduction and clustering analysis of this parcellation illustrated a sparse cluster of projection neurons separated from the remaining subiculum neurons (Fig. 2G). Examining the cells associated with this sparse cluster revealed neurons that selectively targeted the ATN (Fig. 2G, H, Supplementary Fig. 5A), consistent with

**Fig. 1 | Identification of a non-pyramidal excitatory neuron type in the subiculum. A** scRNA-seq landscape of putative excitatory neurons, with cells colored according to cluster identity and visualized via UMAP dimensionality reduction. **B** Expression of control genes (top row) and cluster-specific genes (bottom row) for clusters from (**A**). Results are shown via violin plot, wherein left tick mark denotes zero, and right tick mark denotes maximum value in counts per million (CPM). **C** Cluster-specific mean values of gene expression for all genes. Differentially expressed genes are colored according to their enriched cluster. **D** Atlas schematic of a coronal section of the dorsal subiculum. **E** Expression of *Slc17a7* via chromogenic in situ hybridization. Scale bar: 500 μm. Image from Allen Mouse Brain Atlas[29]. **F, G** As in (**E**), but for the cluster-specific marker genes *Ly6g6e* and *Cck*. **H** Expression of *Slc17a7*, *Ly6g6e*, and *Cck* via FISH, with inset illustrating expansion of shown region. Independently repeated across 7 sections from 3 animals. Scale bars: 100 μm overview, 10 μm inset. **I** Summary of cellular phenotypes for the image in (**H**). Black denotes non-*Slc17a7*-expressing cells. **J** Summary of cellular phenotype

distribution for all excitatory cells examined (n = 142,209 total cells; with n = 73,861 *Slc17a7*-expressing putative excitatory neurons). **K** Combined Nissl and FISH. Arrows denote phenotype of individual cells. Independently repeated across 2 sections from 2 animals. Scale bar: 10 μm. **L** Representative cell-body segmentations in yellow. **M** Cell-body areas for *Ly6g6e*-expressing and *Cck*-expressing cells (n = 301 total cells, with n = 102 and 199 total *Ly6g6e*-expressing and *Cck*-expressing neurons from n = 2 animals; p = 4.3e-2 via two-sided paired t-test on within-animal-averaged data). Data are presented as mean values ± SEM. **N** Schematic illustrating calculation of aspect ratios registered to the alveus for the two cells from (**L**). **O** As in (**M**), but for alveus-oriented aspect ratios (p = 4.3e-2 via two-sided paired t-test on within-animal-averaged data). Data are presented as mean values ± SEM. **P** Principal component analysis of cell-body properties, with points colored according to phenotype from marker gene expression. Cluster 1 (*Ly6g6e*-expressing) phenotype: 116 cells; cluster 2 (*Cck*-expressing) phenotype: 185 cells.

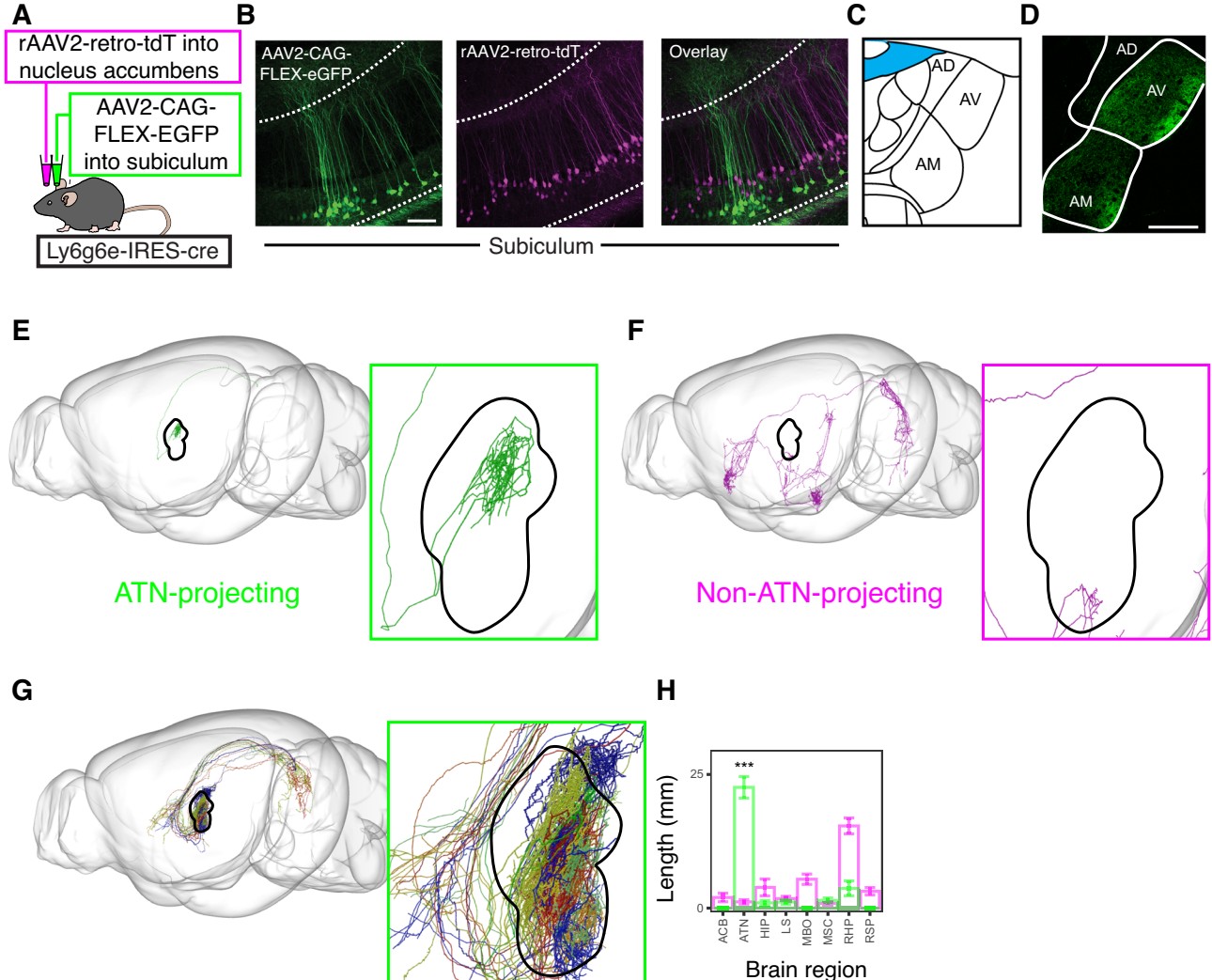

**Fig. 2 | Ovoid neurons exhibit specialized long-range axonal targets.**
**A** Schematic of subtype-specific labeling strategy in *Ly6g6e*-IRES-Cre mouse line.
**B** Representative image of viral expression in the subiculum for strategy in (**A**). Note that images are oriented so that primary apical dendrites of pyramidal cells extend upwards. Independently repeated across 2 sections from 4 animals. Scale bar: 100 μm. **C** Atlas illustrating the anterior thalamic nuclei (ATN).
**D** Representative projections to the anterior thalamic nuclei for CAG-FLEX-EGFP injection scheme shown in (**A**). Scale bar: 300 μm. **E** Axonal reconstruction of a

single subiculum neuron projecting to the ATN. **F** As in (**E**), but for a single neuron with projections elsewhere. **G** Axonal reconstructions for downstream brain regions. **H** Axonal length in downstream brain regions (72 non-ATN-projecting cells, 12 ATN-projecting cells, p = 1.9E-12, two-sided Mann-Whitney U test). Data are presented as mean values ± SEM. ACB Nucleus Accumbens, ATN Anterior Thalamic Nuclei, HIP Hippocampal Region, LS Lateral Septal Nucleus, MBO Mammillary Body, MSC Medial Septal Complex, RHP Retrohippocampal Region, RSP Retrosplenial Area.

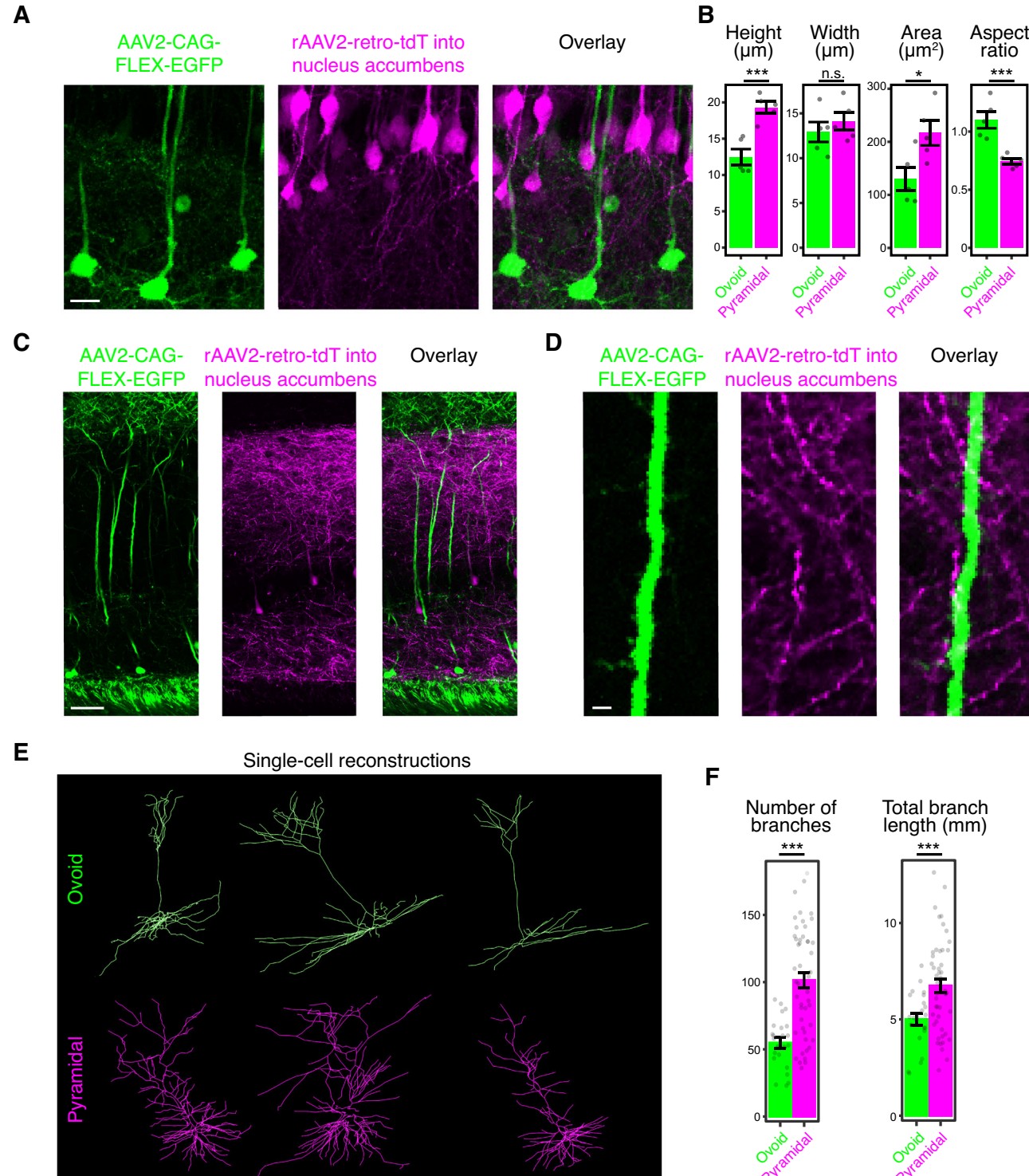

**Fig. 3 | Ovoid neurons have specialized dendritic architectures.**
**A** Representative subiculum cell bodies for neurons labeled by the viral scheme in Fig. 2A; i.e., by Cre-dependent eGFP into Ly6g6e-cre mouse (green) along via retrograde-labeled neurons projecting to the nucleus accumbens (magenta). Scale bar: 20 μm. **B** Summary of cell-body properties for neurons labeled by scheme in (**A**). Data points are averages of individual animals (n = 57 ovoid cells and n = 58 pyramidal cells from n = 5 animals for each cell type, width: *p* = 4.2e-1; height: *p* = 7.9e-3; area: 3.2e-2; aspect ratio: *p* = 7.9e-3, two-sided Mann-Whitney U test). Data are presented as mean values ± SEM. **C** Representative image of labeled neurons for the viral scheme shown in (**A**). Scale bar: 50 μm. **D** Expansion around primary apical dendrite of an ovoid neuron. Scale bar: 2 μm. **E** Representative single-cell morphologies of ATN-projecting putative ovoid neurons (green, top row) and nucleus accumbens-projecting pyramidal neurons (magenta, bottom row). Morphologies obtained via Janelia Mouselight Project[35]. **F** Dendritic properties of single-cell ATN-projecting putative ovoid neurons (green, top row) and nucleus accumbens-projecting putative pyramidal neurons (magenta, bottom row). Data points are individual neurons (number of branches: *p* = 6.8e-9; length: *p* = 3.6e-4, n = 23 ovoid cells and n = 48 pyramidal cells, two-sided Mann-Whitney U test). Data are presented as mean values ± SEM. Morphologies obtained via Janelia Mouselight Project[35].

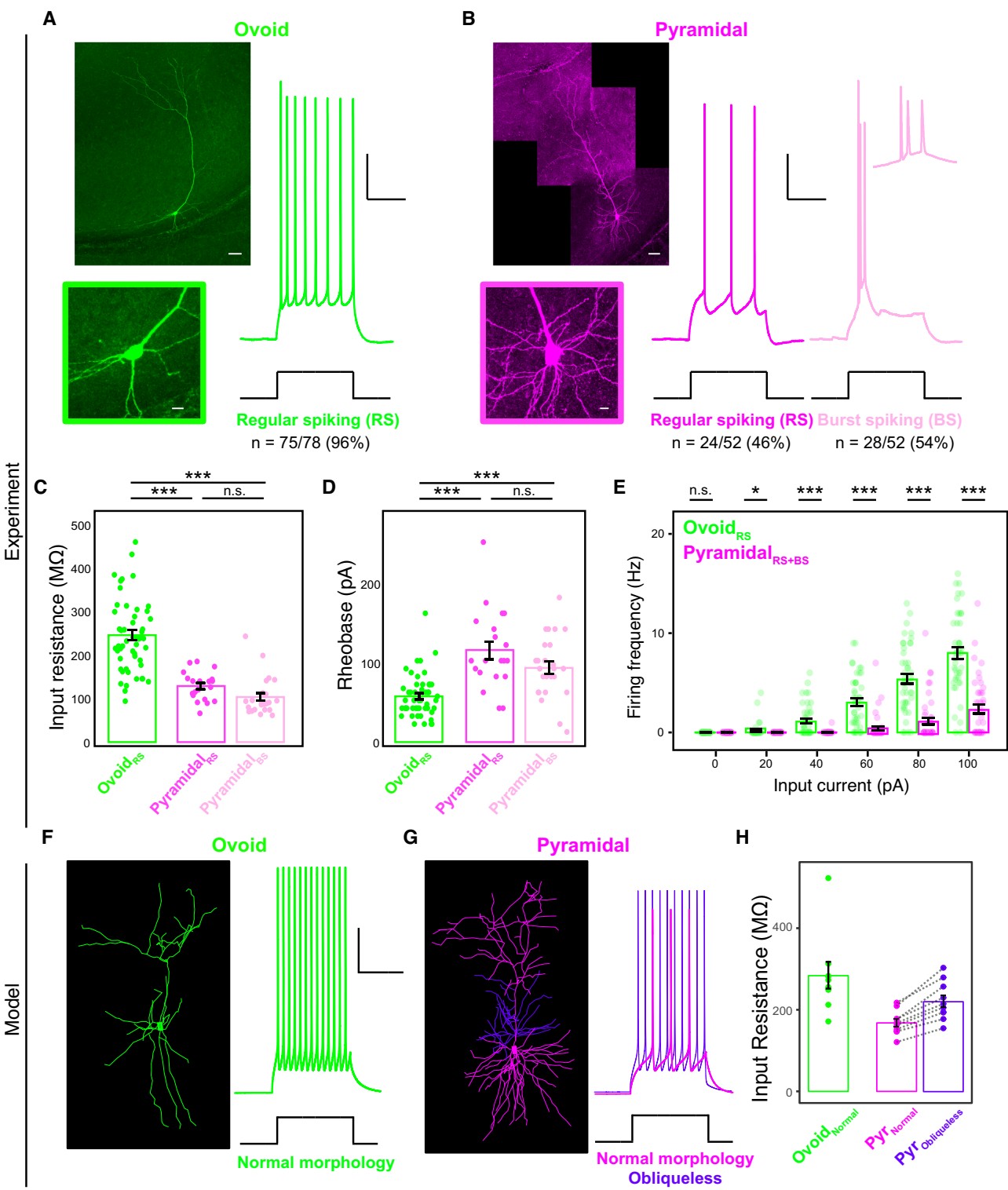

distinct projection targets of ovoid neurons and pyramidal neurons. In combination, our bulk labeling illustrates a singular projection from *Ly6g6e*-expressing ovoid neurons to the ATN, which can be uniquely explained by a specialized ATN-projecting cluster within single-cell projection data. Such a relationship illustrates ovoid neuron-specific ATN projections across several complementary datasets.

**Ovoid neurons have specialized dendritic arbors**
Given that ovoid neurons exhibited a specialized spatial location (Fig. 1I, J), cell-body morphology (Fig. 1K–P), and projection pattern

(Fig. 2A–I), we were next motivated to assess whether their dendritic morphology was similarly specialized. First, we ensured that viral labeling (via the labeling strategy of Fig. 2A) exhibited cell-body properties consistent with ovoid and pyramidal neurons (Fig. 3A, B), confirming the predictions from our Nissl-DAPI-based analysis (cf. Fig. 1L–P). Examining virus-labeled dendritic morphology, we noted that ovoid neurons typically lacked radial oblique dendrites, a hallmark of classical pyramidal neurons (Fig. 3C, D; Supplementary Movie 2). Analyzing single-cell dendritic properties of ATN-projecting neurons (via http://mouselight.janelia.org: ref. 37) confirmed this markedly

**Fig. 4 | Ovoid neurons have heightened excitability, consistent with specialized morphology. A** Left: Ovoid cell morphology from ex vivo whole-cell recording. Inset provides expanded cell body. **B** As in (**A**), but for pyramidal cells. Inset shows bursting. **C** Input resistances for ovoid neurons, regular spiking pyramidal neurons, and burst spiking pyramidal neurons ($n = 75$ ovoid cells, $n = 24$ regular spiking pyramidal cells, $n = 28$ burst spiking pyramidal cells, $n = $ total 67 animals; ovoid vs. regular spiking pyramidal $p = 6.1e\text{-}8$, ovoid vs. burst spiking pyramidal $p = 6.1e\text{-}8$, regular spiking pyramidal vs. burst spiking pyramidal $p = 5.9e\text{-}3$, Kruskal-Wallis test on averaged data from individual animals). Data points are averages from individual animals and data are presented as mean values ± SEM. **D** As in (**C**), but for rheobase of cells (ovoid vs. pyramidal regular spiking $p = 1.9e\text{-}6$, ovoid vs. burst spiking pyramidal $p = 3.9e\text{-}5$, regular spiking pyramidal vs. burst spiking pyramidal $p = 1.4e\text{-}1$, Kruskal-Wallis test). **E** Input-output curves for ovoid and pyramidal neurons, with

regular spiking and bursting neurons pooled for pyramidal neurons ($n = 71$ ovoid, $n = 70$ pyramidal cells, $n = 67$ animals; 0pA $p = 1$, 20pA $p = 2.4e\text{-}2$, 40pA $p = 2.7e\text{-}7$, 60pA $p = 9.7e\text{-}10$, 80pA $p = 1.0e\text{-}9$, 100pA $p = 1.1e\text{-}10$ on averaged data from individual animals, two-sided Mann-Whitney U test). Data are presented as mean values ± SEM. **F** Left: Reconstructed morphology of an ovoid neuron used for simulation. Right: action potential firing for ovoid neuron following current injection. Scale bars: 20 mV and 250 ms. **G** As in (**F**), but for a pyramidal neuron, with radial oblique dendrites colored in violet. Overlaid traces depict simulations with radial obliques ("normal morphology") and without radial obliques ("obliqueless"). **H** Input resistances for modeled ovoid neurons ($n = 9$), pyramidal neurons with radial obliques ("normal morphology", $n = 10$), and pyramidal neurons without radial oblique dendrites ("obliqueless", $n = 10$). Data points are individual modeled neurons and data are presented as mean values ± SEM.

different dendritic morphology (Fig. 3E), wherein ATN-projecting putative ovoid neurons were characterized by fewer branches and reduced overall branch length in (Fig. 3F). Thus, multiple complementary datasets are consistent with dendritic variation between ovoid neurons and pyramidal neurons.

## Ovoid neurons are hyperexcitable relative to pyramidal cells

The distinct structural properties of ovoid neurons suggest that these cells may have unique functional properties. As previous literature has shown that the subiculum contains a mix of regular spiking and bursting neurons ex vivo[24], we sought to determine whether ovoid neurons would have subtype-specific firing patterns, and examined this through whole-cell recordings from both ovoid and pyramidal neurons in brain slices. Strikingly, nearly all recorded ovoid neurons exhibited regular firing properties ($n = 75/78$ cells; Fig. 4A), whereas pyramidal cells exhibited a mixture of regular spiking and burst spiking ($n = 24/52$ cells and $n = 28/52$ cells respectively; Fig. 4B). As expected, morphological detection performed *post hoc* reinforced the variation between ovoid and pyramidal neurons in both cell-body geometry (Supplementary Fig. 5B, C) and dendritic arbors (Fig. 4A, B).

Consistent with their lack of radial obliques and reduced overall length, ovoid neurons showed a markedly higher input resistance relative to pyramidal neurons (Fig. 4C; no effect of genotype: Supplementary Fig. 5D). Ovoid neurons also exhibited significant differences in electrophysiological properties contributing to excitability, including lower rheobase and a steeper input-output curve (Fig. 4D, E; no difference in resting membrane potential: Supplementary Fig. 5E; difference in sag ratio: Supplementary Fig. 5F). Computational modeling confirmed that the lack of radial obliques in ovoid cells was sufficient to drive markedly higher input resistance (Fig. 4F, G), and that removal of radial obliques helped to increase input resistance in pyramidal cells (Fig. 4H). These collective results show that, consistent with their distinctive morphology, ovoid cells have heightened intrinsic excitability.

## Slow, novelty-driven responses specific to ovoid neurons

To assess if ovoid neuron activity was also specialized relative to pyramidal neuron activity in vivo, we next performed calcium imaging using wireless 1-photon miniaturized fluorescence microscopes[38]. To image ovoid neurons, we injected a Cre-dependent calcium indicator (AAV-Syn-FLEX-GCaMP6f) in our Ly6g6e-IRES-cre mouse line and compared activity to broad subiculum activity assayed in a non-cell-type-specific manner (via AAV-Syn-GCaMP6f, presumably strongly biased to assessing pyramidal cell activity due to their high abundance relative to ovoid cells) (Fig. 5A, B). This imaging revealed that ovoid neurons had slow, spontaneous, and synchronized activity relative to that of pyramidal neurons (example activity: Fig. 5C; summary: Supplementary Fig. 6B–D).

Subiculum-to-ATN projections form the initial component of the Papez circuitry, which plays critical roles in memory and attention[23,39]. To examine whether there would be ovoid neuron-specific

contributions to these roles, we trained and tested animals in an object recognition paradigm, which might engage ovoid neurons through learning and memory and/or novelty-triggered attentional states. In our paradigm, memory encoding was induced by allowing mice to explore two identical novel objects for 5 min, and after variable delays (1–100 days), memory retrieval was examined by exposing the animal to one previously experienced object and one novel object (Fig. 5D). Imaging was performed throughout encoding and retrieval, with cellular activity z-scored and examined relative to object interactions. Remarkably, ovoid cells exhibited sustained population activity correlating with novel object encounters, but lacked similar activity when encountering familiar objects (average across all cells from representative animal: Fig. 5E, example trials: Supplementary Fig. 6E, F; example ΔF/F: Supplementary Fig. 6G, H; results robust to ovoid cell targeting strategy: Supplementary Fig. 7A–L). In contrast, such sustained activity was absent in pyramidal cells to either novel or familiar object encounters (Fig. 5F).

To quantify cells responding to objects, we categorized cells into "responders" and "non-responders" to local objects, with cells considered a "responder" if their activity exceeded a minimum of 0.25 ΔZ for a minimum of 10 s (Fig. 5G; responder categorization cross-validation: Supplementary Fig. 6I–L). Using this categorization of cellular activity, the majority of ovoid cells were responsive to novel objects when considered over the full extent of behavioral session, with the ensemble responding to novel objects exhibiting some fluidity within the behavioral session (Supplementary Fig. 6K, L). In contrast, ovoid neurons exhibited little responsiveness to familiar objects (Fig. 5E). This cellular expression of memory for previously encountered objects extended to objects experienced months in the past (Fig. 5H), and outlasted the time window of behavioral expression of object memory (Supplementary Fig. 6N).

In contrast to ovoid cells, few pyramidal cells met the criteria for a responder cell to either novel or familiar objects (Fig. 5H), which held across a range of responder criteria durations (Supplementary Fig. 6O). Selective recruitment of ovoid neurons was recapitulated by a duration-parameter-free approach that examined cellular activity peaks agnostic to decay kinetics ($p < 0.001$ for ovoid vs. pyramidal neuron percentile-normalized peak activity; Supplementary Fig. 6O, see "Methods"), as well as analysis of fluorescence changes relative to pre-encounter baseline activity (ΔF/F analysis: Supplementary Fig. 6P). Finally, slow novel-object-triggered responses in ovoid neurons were robust to a variety of viral and transgenic targeting strategies (Supplementary Fig. 7A–L).

In contrast to novel object-induced activity, ovoid neurons generally did not respond to novel spatial locations of familiar objects (Supplementary Fig. 8A–F), illustrating that ovoid neuron activity was selective for a non-spatial form of novelty. Within our behavioral assays, ovoid neurons exhibited minimal place tuning (Supplementary Fig. 8G; consistent with low levels of sharp place-field tuning of subiculum neurons assessed via calcium imaging[40]). Ovoid neurons also lacked strong tuning to speed (Supplementary Fig. 8H) and general

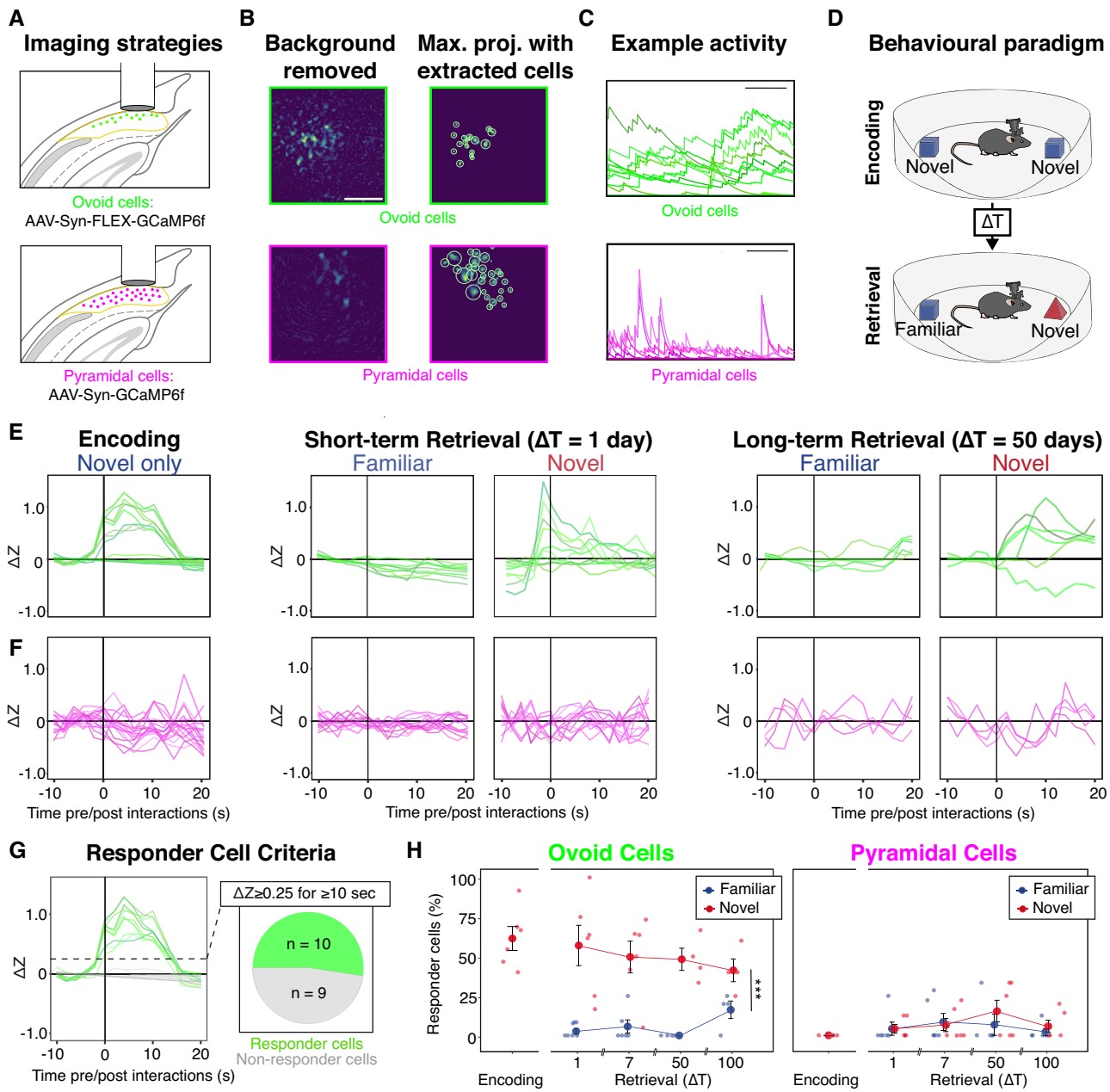

**Fig. 5 | Selective responses to novel objects in ovoid cells. A** Viral and imaging strategy for assessing activity in ovoid neurons (top) and pyramidal neurons (bottom) with 1-photon calcium imaging, with approximate GRIN lens placement. **B** Example fields of view with background-removed activity maps (left) as well as maximum projections with extracted segmented cells (right), for both ovoid cells (top row) and pyramidal cells (bottom row). Scale bar: 100 μm. This was independently repeated 12 times. **C** Example calcium traces for ovoid neurons (top) and pyramidal neurons (bottom) taken during exploration in the novel object-encoding session. Scale bar: 10 s, with y-axis having arbitrary units. **D** Schematic of novel object recognition paradigm. **E** Object-interaction-triggered ovoid neuron activity for novel and familiar object interactions, for a representative animal undergoing the behavioral paradigm illustrated in (**D**). Vertical black line at 0 denotes start of interaction with an object, with individual lines illustrating averaged activity of all segmented individual cells within the field of view for the session. **F** As in (**E**), but for a representative animal assessing pyramidal cell activity. **G** Depiction of classification of responder cells, wherein cells must reach a minimum ΔZ of 0.25 for a minimum of 10 s following object interactions. **H** The percentage of cells that classify as a responder cell for novel (red) and familiar (blue) objects, for ovoid cells (left) and pyramidal cells (right). Data points represent individual animals (n = 6 animals for each of ovoid and pyramidal conditions, ovoid: p = 3.1e-3, pyramidal: p = 0.48, two-sided linear model, no adjustment for multiple comparisons) and data are presented as mean values ± SEM.

mobility (Supplementary Fig. 8I) (consistent with previous reports of speed-tuned cells being enriched in retrosplenial-cortex-projecting subiculum distal to our recording site[12]). Further reinforcing the lack of spatial representations in ovoid cells, changing the spatial context between novel object recognition sessions (via abrupt lighting changes or changes to overall spatial environments) failed to evoke responses in ovoid neurons. Supplementary Fig. 8J–N). In collection, this suite of

experiments converged upon a long-term and non-spatial novelty phenotype specific to ovoid neurons.

**Inhibition of ovoid neurons prevents memory encoding of objects**

To examine whether ovoid neurons play a necessary role in object memory, we used ArchT-based optogenetic inhibition to silence ovoid

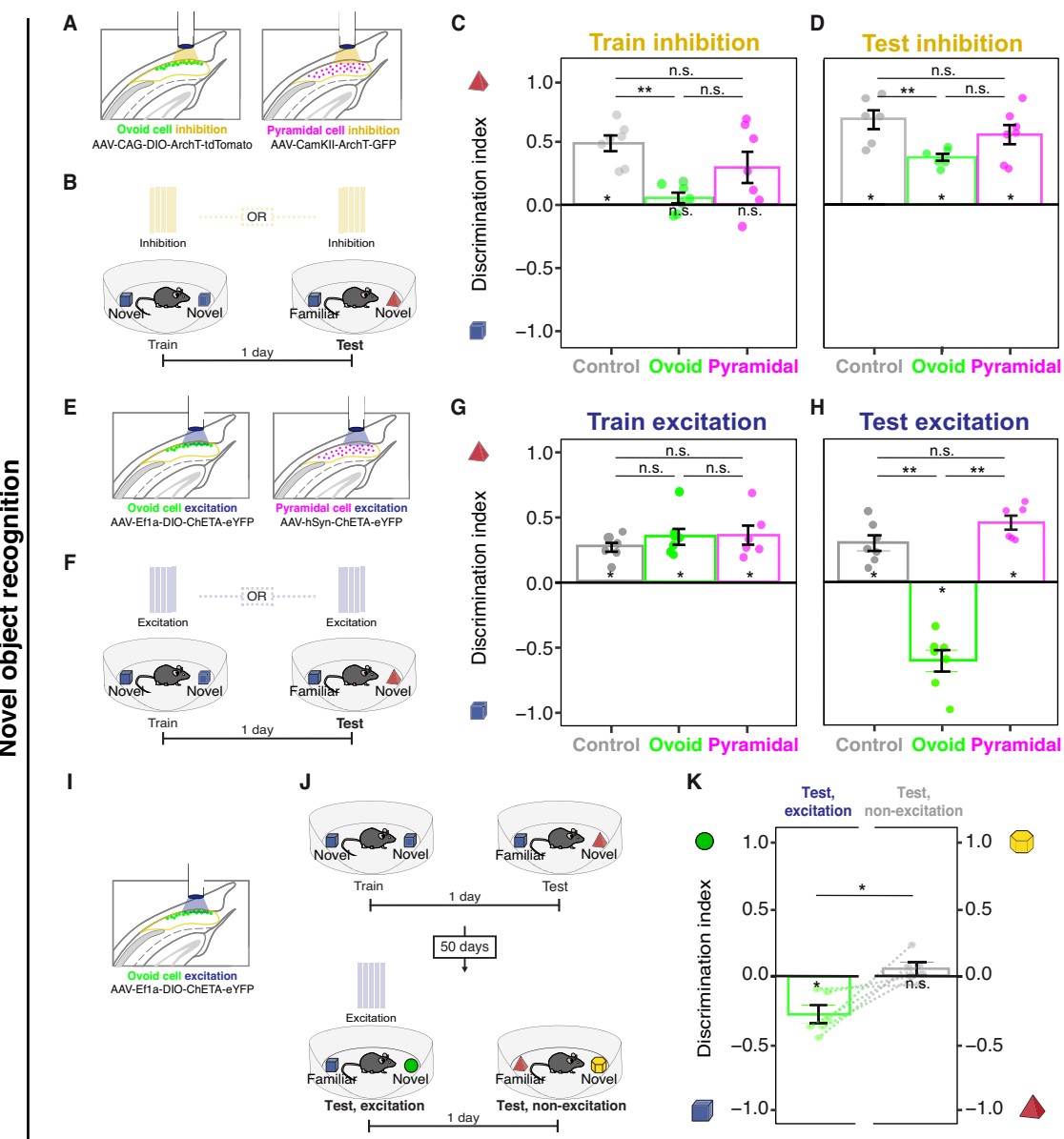

**Fig. 6 | Optogenetic manipulation of ovoid cells toggles object preference.**
**A** Viral strategy for targeting ovoid (left) and pyramidal cells (right). **B** Novel object recognition and optogenetic inhibition. **C** Discrimination indices on test day for inhibition occurring during training session (n = 7 control, n = 7 ovoid, n = 7 pyramidal; ovoid vs. pyramidal p = 0.25, ovoid vs. control p = 0.012, pyramidal vs. control p = 0.06, Kruskal-Wallis test; training inhibition from 0, control p = 0.031, ovoid p = 0.29, pyramidal p = 0.078, two-sided Mann-Whitney U test). Data are presented as mean values ± SEM. **D** As in (**C**), but for inhibition during testing (n = 7 control, n = 6 ovoid, 7 pyramidal; ovoid vs. pyramidal p = 0.20, ovoid vs. control p = 8.2e-3, pyramidal vs. control p = 0.20, Kruskal-Wallis test; test inhibition from 0, control p = 0.031, ovoid p = 0.031, pyramidal p = 0.016, two-sided Mann-Whitney U test). **E**, **F** As in (**A**, **B**), but for ChETA-based ovoid neuron excitation during training, assessing encoding. **G**, **H** As in (**C**, **D**), but for excitation during testing, assessing

short-term retrieval (n = 7 control, n = 7 ovoid, n = 6 pyramidal; Training excitation: ovoid vs. control p = 0.82, ovoid vs. pyramidal p = 0.44, control vs. pyramidal p = 0.72; train excitation from 0, control p = 0.02, ovoid p = 7.8e-3, pyramidal p = 0.022. Test excitation: ovoid vs. control p = 1.5e-2, ovoid vs. pyramidal p = 2.4e-3, control vs. pyramidal p = 0.35, Kruskal-Wallis test; test excitation from 0, control p = 0.031, ovoid p = 7.8e-3, pyramidal p = 0.031, two-sided Mann-Whitney U test). **I**, **J** As in (**E**, **F**), but for excitation assessing long-term retrieval, occurring 50 days after initial encoding, as well as a subsequent non-excitation retrieval test. **K** Discrimination indices for preferences during stimulation test day and non-stimulation test day for long-term retrieval (n = 6 control, n = 6 ovoid; p = 8.7e-3, Kruskal-Wallis test; test excitation from 0, test excitation p = 0.031, test non-excitation p = 0.44, two-sided Mann-Whitney U test). Data are presented as mean values ± SEM.

neurons (via AAV-CAG-DIO-ArchT-tdTomato in Ly6g6e-IRES-cre mice). In a given mouse, light-based inhibition of ovoid neurons occurred during encoding or short-term retrieval (1-day delay) in the novel object assay (Fig. 6A, B). Object preference was assessed on retrieval days by calculating the discrimination index between novel and familiar objects (defined as the time difference spent investigating novel and familiar objects, divided by the total object investigation time), and compared to control mice injected solely with an AAV-CAG-GFP virus.

Optogenetic inhibition of ovoid neurons produced marked encoding deficits relative to controls, such that discrimination was reduced to chance levels during subsequent retrieval (Fig. 6C). Conversely, inhibition during the retrieval session produced a relatively modest discrimination impairment that remained above chance levels (Fig. 6D), highlighting an indispensable role for ovoid neurons selectively in memory encoding. This encoding phenotype was not attributable to overall differences in object exploration, as inhibition had no discernable effects on exploration during either encoding or

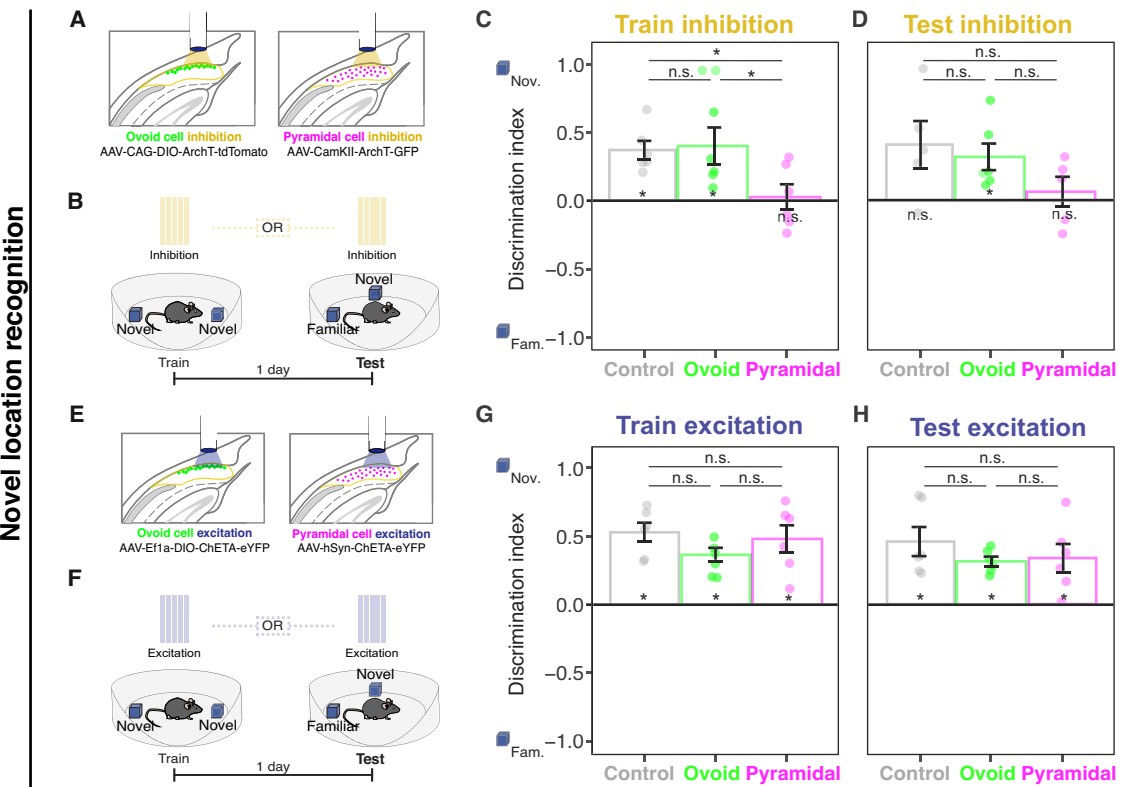

**Fig. 7 | Optogenetic inhibition of pyramidal cells impairs spatial learning while ovoid inhibition spares spatial learning. A** Viral strategy for targeting ovoid (left) and pyramidal cells (right). **B** Novel location recognition and optogenetic paradigm for ArchT-based inhibition. **C** Discrimination indices on test day for inhibition occurring during training session (n = 6 control, n = 7 ovoid, n = 6 pyramidal; Training inhibition: pyramidal vs. ovoid p = 0.045, pyramidal vs. control p = 0.031, ovoid vs. control p = 0.81, Kruskal-Wallis test; training inhibition from 0, controls p = 0.031, ovoid p = 0.031, pyramidal p = 0.56, two-sided Mann-Whitney U test). Data are presented as mean values ± SEM. **D** As in (**C**), but for inhibition during testing (n = 6 control, n = 7 ovoid, n = 6 pyramidal; test inhibition: pyramidal vs. ovoid p = 0.41, pyramidal vs. control p = 0.095, ovoid vs. control p = 0.52, Kruskal-

Wallis test; test inhibition from 0, control p = 0.13, ovoid p-value p = 0.031, pyramidal p = 0.63, two-sided Mann-Whitney U test). Data are presented as mean values ± SEM. **E–H** as in (**A–D**) but for the ChETA-based ovoid neuron excitation during novel location recognition (n = 6 control, n = 7 ovoid, n = 6 pyramidal; Training excitation: control vs. ovoid p = 0.066, control vs. pyramidal p = 0.81, ovoid vs. pyramidal p = 0.23, Kruskal-Wallis test; train excitation from 0, control p = 0.031, ovoid p = 0.031, pyramidal p = 0.031, two-sided Mann-Whitney U test. Test excitation: control vs. ovoid p = 0.52, control vs. pyramidal p = 0.58, ovoid vs. pyramidal p = 1, Kruskal-Wallis test; test excitation from 0, control p = 0.031, ovoid p = 0.031, pyramidal p = 0.031, two-sided Mann-Whitney U test). Data are presented as mean values ± SEM.

retrieval (Supplementary Fig. 9A–D). Remarkably, silencing ovoid neurons had no effect in either the encoding or retrieval of object locations (Fig. 7A–D), illustrating that ovoid neuron deficits were selective for object memory relative to location memory. Additionally, the role of ovoid neurons could be functionally dissociated from pyramidal cells: silencing pyramidal cells in this experiment evoked both encoding and retrieval deficits in spatial location assay (Fig. 7A–D), further highlighting a subtype-specific non-spatial role for ovoid neurons.

**Activation of ovoid neurons evokes familiarity seeking**
To further investigate a causal role of ovoid cells in novel-object-related behavior, we next used ChR2-based optogenetic activation targeted to ovoid neurons (via AAV-Ef1a-DIO-ChETA-EYFP in Ly6g6e-IRES-cre mice). In this paradigm, a 10-s-on/20-s-off light cycle was used to mimic the slow timescales of activity seen from imaging (cf. Fig. 5C, Supplementary Fig. 6B–D; qualitatively identical results received with 2-s-on stimulation: Supplementary Fig. 9E, F).

When excitation was performed during encoding (Fig. 6E, F), mice showed novel object learning, with no significant effect from ovoid neuron stimulation (Fig. 6G). Conversely, when excitation occurred during short-term retrieval (1-day delay: Fig. 6F), ovoid neuron activation caused a preference for the familiar object (Fig. 6H). For

long-term retrieval (50-day delay after encoding: Fig. 6I, J), ovoid neuron stimulation was sufficient to drive a significant bias for the familiar object (blue: Fig. 6K), despite animals exhibiting behavioral extinction of object memory when lacking stimulation (gray: Fig. 6K; see also Supplementary Fig. 6N).

This familiarity bias during retrieval could not be explained by novelty aversion, as light stimulation did not drive novelty aversion during encoding, nor could it be explained by overall differences in total object exploration (Supplementary Fig. 9G–L). In contrast to the object recognition task, activation of ovoid neurons during the spatial location task produced no familiarity bias (Fig. 7E–H), illustrating that familiarity seeking was selective for a non-spatial task. Intriguingly, stimulation produced no change in novelty preference in pyramidal neurons (Fig. 6G, H), illustrating an ovoid neuron-specific effect. In collection, these results illustrate that when previously encountered objects are present, ovoid neuron activation is sufficient to drive familiarity seeking of both recently and remotely experienced objects, and this behavior can be dissociated from pyramidal neurons. This pronounced phenotype highlights the specialized role that ovoid neurons play in object recognition, and as behavioral phenotypes were similar in light-on and light-off periods (Supplementary Fig. 9H, J, L), suggests that ovoid neuron manipulations may control slow brain states.

## Discussion

### Ovoid neurons and non-spatial novelty representation

Pyramidal neurons of the hippocampus generate representations of spatial and non-spatial content[1-6]. However, it is unclear whether these representations reflect a spatial network that embeds non-spatial information, or whether non-spatial content can be dissociated from spatial content[7-10]. Notably, these representational properties and inferred relationships are frequently studied from the perspective of CA1 pyramidal cells, which have a variety of anatomically and functionally distinct properties relative to the subiculum[11,22]. Indeed, comparative lesion studies suggest that the subiculum may be more important for non-spatial information relative to CA1[41,42]. Given the existence of discrete subtypes of subiculum excitatory neurons[25,26] that are not found in CA1[43], the subiculum provides a key opportunity to decipher the subtype-specific logic of hippocampal representations and contributions to function.

Motivated by this, we identified a transcriptomically outlying and spatially well-defined subtype of excitatory neurons in the subiculum (Fig. 1) and sought to map higher-order properties and representations of this neuronal subtype. In doing so, we first identified that this subtype was a non-pyramidal "ovoid" cell, which appeared to be structurally and functionally poised to perform different computations relative to classical pyramidal neurons (Figs. 2–4). Consistent with this prediction, we found ovoid neurons were experientially tuned, such that they responded to novel but not familiar objects (Fig. 5) and made distinct contributions to memory from pyramidal neurons when probed causally (Figs. 6 and 7; Supplementary Fig. 9). Notably, this responsiveness was selective for non-spatial features, as ovoid cells did not show significant responses to familiar objects in novel locations nor to novel environmental changes (Supplementary Fig. 8). Further, when probed with optogenetic inhibition and activation, only pyramidal cells showed altered responses during spatial novelty assays (Fig. 7). These findings illustrate fundamental functional differences between ovoid neurons and pyramidal neurons, reflecting a cell-type-specific double dissociation of two of the classical roles of the hippocampus.

### The computational role of ovoid neurons in object memory

What is the functional role of ovoid neuron activity in object recognition memory? Given that ovoid neurons respond to novel objects, but lack responses to familiar objects (Fig. 5), these correlative imaging results would suggest a selective role in the encoding of objects into memory, but have a negligible effect during subsequent retrieval. Such a prediction was recapitulated by our inhibition experiments (Fig. 6A–D), wherein silencing ovoid neurons produced a pronounced object memory encoding deficit, illustrating that ovoid neurons play a necessary role in memory encoding.

We note that this does not argue for ovoid neuron encoding the specific content of novel objects per se; indeed, there are many features of ovoid neuron responses that are consistent with brain-state effects that permit memory encoding broadly, rather than instructing precise elements of memory for individual objects. Critically, the large number of ovoid cells physiologically activated by novel objects (Fig. 5H) is suggestive of a coarse novelty signal, rather than one conveying precise information about the particular novel object stimulus. Additionally, optogenetic manipulation of ovoid neurons shows similar behavior states in light-on and light-off periods (Supplementary Fig. 9D, J, L), providing further evidence for slow changes to brain state that span beyond instantaneous perturbations. Finally, ovoid neurons responding to novel objects change throughout a session (Supplementary Fig. 6K, L), illustrating that ovoid neurons likely signal via an ensemble code. Altogether, these findings suggest that ovoid neurons relay a coarse novelty signal in the brain that may permit the operation of complex downstream memory systems.

Given these experimental results, one computational role for ovoid neurons may be to act as a novelty-associated gate, permitting a downstream memory network to become accessible and updateable with specific information on objects. In such a role, object novelty would activate ovoid neurons, with ovoid activity permitting a downstream brain region to encode new object-specific information conveyed via other circuits[44]. This putative computational role is consistent with our imaging results, wherein object novelty triggered slow brain-state-like responses in ovoid neurons (Fig. 5). This is role also in agreement with both our ovoid neuron silencing and activation experiments. Specifically, this role would predict that silencing ovoid neurons would abolish their putative permissive role in memory updating, consistent with the encoding deficit seen in our inhibition experiments (Fig. 6C). Additionally, this role would predict that selective activation of ovoid neurons could impart strong drive to this downstream memory network, and bias brain state towards the retrieval of previously encountered objects rather than the encoding of new objects. This can account for the behavioral expression of familiar-object seeking we see in our activation experiments (Fig. 6H), and is akin to behavioral place-seeking following activation of pyramidal cells that correspond to a given place[45].

### Mechanisms of the object novelty response

Mechanistically, we propose that novelty responses in ovoid neurons reflect the combination of instantaneous sensory input from the entorhinal cortex[9], occurring in temporal conjunction with known novelty-associated acetylcholine signals from the medial septum[46,47]. As $Ly6g6e$ has been shown to potentiate cholinergic drive by slowing acetylcholine receptor desensitization[48], this interaction is consistent with the slow novelty-triggered ovoid neuron activity in calcium imaging recordings (Fig. 5). These effects may be further compounded by cholinergic modulation of local inhibitory neurons[49,50], although the extent to which ovoid neurons are wired into cell-type-specific local networks has yet to be resolved.

Heightened novel-object-induced activity from ovoid neurons likely provides a strong drive for synaptic long-term potentiation in the ATN (Fig. 2), and promotes memory consolidation of experienced objects. This hypothesis is consistent with recently published work illustrating the hippocampus-ATN-prefrontal cortex circuit governing the stabilization of long-term memories[51], and is supported here by ovoid-specific novel object recognition deficits after inhibition of the encoding of objects (Fig. 6). Additionally, artificial reactivation of ovoid neurons (Figs. 6 and 7) may re-evoke the memory trace of these experienced objects, and drive our observed behavioral expression of familiarity seeking. In collection, these proposed mechanisms would unite specialized ovoid neuron molecular, cellular, and circuit properties (Figs. 1–4), and help explain how these specializations contribute to subtype-specific feature selectivity (Fig. 5) and the control of behavioral preference (Figs. 6 and 7).

### Comparison and extensions to previous studies

Our work here complements an existing body of work studying structural and functional properties of the hippocampus. Structurally, previous work has illustrated that the deep subiculum selectively projects to the ATN[34-36]. Our work here shows that this projection emerges from ovoid neurons that are transcriptomically, morphologically, and physiologically specialized relative to pyramidal cells. Although our findings here illustrates that the ATN receives its subiculum input from ovoid neurons, it should be noted that our work does not necessarily imply that all ovoid neurons project to the ATN. Through our long-range circuit mapping, we can exclude the possibility of ovoid neurons projecting to long-range targets outside of the ATN (Fig. 2A–D), but there remains the possibility that some ovoid neurons may have solely local projections within the hippocampus.

Functionally, a variety of work has examined object-coding properties of hippocampal neurons, including excitatory neurons in the subiculum[9,20,21]. Remarkably, using multi-electrode recordings across CA1 and subiculum, one such study illustrated a small subset of cells selectively responded to novel objects, lacked apparent spatial tuning, and that these cells comprised a sparse subset of cells found in the subiculum[20]. From this similarity, such neurons may be the ovoid neurons characterized here. Other work, using spatially resolved assays, has shown that cells can have object-encoding properties that are not necessarily consistent with the deep spatial location of ovoid neurons[52] or operate on different timescales[21]. It is likely that some of these cells are captured in our pyramidal cell imaging datasets, as a small, but non-zero, number of cells respond to objects under the temporally slow responder classification we have employed (Fig. 5G), and note that additional object responsiveness is likely under more lenient classification schemes. Thus, it is likely that ovoid neurons govern a specialized novel-object-driven activity in the subiculum, which is complemented by a variety of other object-dependent feature selectivity properties from pyramidal neurons.

Our work also extends these previous studies, attributing a variety of results to the ovoid cell type and differentiating their contribution to memory relative to classical pyramidal cells. Specifically, we have shown that inhibition of ovoid neurons during encoding caused an impairment in non-spatial object learning (Fig. 6C, D), while the reactivation of ovoid neurons was sufficient to drive behavioral bias to experienced objects for recent memory (Fig. 6G, H) as well as remote memory that was behaviorally latent (Fig. 6K). Remarkably, all of these phenotypes were dissociable relative to pyramidal cells. Importantly, this cellular expression of memory in ovoid neurons varies from classical pyramidal neuron studies and engram frameworks for memory[53], which is predicated upon neurons being active during both encoding and retrieval conditions. In collection, ovoid neurons encapsulate a new form of cellular expression of memory, wherein single brief learned experiences can produce sustained changes in neural activity for months (Fig. 5), and months later artificial reactivation can drive behavioral expression of those experiences (Fig. 6).

In our imaging experiments, both ovoid and pyramidal neurons showed little place-field tuning (Fig. S8G). This is in contrast to other studies, which have revealed spatially diffuse tuning in subiculum neurons (e.g., refs. 13,17,18). Such differences are likely attributable to at least two technical reasons. One, in previous studies spatial tuning has typically been assayed in electrophysiological recordings, whereas the slower calcium imaging used in our study may distort temporal relationships between cellular activity and animal location[54]. Consistent with this, other work that imaged subiculum neurons using a 1-photon miniscope has also reported little place-field tuning[38]. Two, we used a large circular arena and brief behavioral experiments to facilitate our object recognition assays, whereas smaller non-circular arenas and longer behavioral sessions may be more advantageous for detecting spatial tuning[12,15–18]. As a consequence, our interventional experiments may act as a better paradigm for assessing importance of ovoid and pyramidal neurons in spatial computations. These experiments showed dissociable contributions between ovoid and pyramidal neurons, such that ovoid neuron activity was not required for the behavioral expression of memory in a spatial location assay, whereas pyramidal neuron activity was necessary for both memory encoding and retrieval (Fig. 7).

### The subiculum is layered in cell types and computation

Pyramidal neurons are conventionally considered to be the sole excitatory neuron type in the subiculum[55]. Here, we examined a transcriptomically unique excitatory neuronal subtype, and spatially mapped this subtype to the deepest region of the subiculum (Fig. 1) (see also refs. 25,26). This coarse spatial location has been studied in non-human primates and humans and has been referred to as the "polymorphic layer"[33,34,56]. This layer is named for its morphologically distinct cells, which are known to include specialized inhibitory neurons[49]; here, we demonstrate the surprising finding that excitatory non-pyramidal neurons are a primary contributor to the polymorphism of this layer. Intriguingly, prior work analyzing human gene expression via in situ hybridization has identified sparse, deep subicular cells that exhibit many of the marker genes found here (e.g., *Chrm2*: Supplementary Fig. 1C)[49,50]. These findings suggest that ovoid cells are likely present in the human subiculum.

Excitatory ovoid neurons within the polymorphic layer likely add critical computational complexity to the hippocampus and the brain. Within the hippocampus, the spatial adjacency of ovoid neurons in the polymorphic layer and nearby pyramidal neurons (Fig. 1) may facilitate synaptic interactions between these two types of neurons, and potentially embed novelty-associated information from ovoid neurons into more classical representations of the spatial environment in pyramidal cells. Beyond the hippocampus, the slow activity observed here in ATN-projecting ovoid neurons is in agreement with slow ATN timescales observed experimentally[51], and it is likely that novelty-related information from ovoid neurons provides important drive to the Papez circuit via this projection[57]. Given the key non-spatial role of the Papez circuit in attention[57], such ovoid-relayed novelty signals would be consistent with the known behavioral coupling between novelty and attention[58]. As the polymorphic layer is structurally conserved across rodents, non-human primates, and humans, it is likely that ovoid neurons broadly play key roles in cognition.

## Methods

### Animals

Experiments used adult (minimum 6 weeks of age) mice of both sexes, maintained on a 12-h light-dark cycle with *ad libitum* access to food and water. All experimental procedures involving mice were approved by the University of British Columbia Animal Care Committee and/or the Janelia Research Campus Institutional Animal Care and Use Committee, as applicable.

### Analysis of single-cell RNA-seq data

Previously published data[25] was used for scRNA-seq analysis. Computational analysis was performed in R (RRID:SCR_001905)[59] using a combination of Seurat v4 (RRID:SCR_007322)[60,61] and custom scripts[24]. Non-neuronal cells and interneurons were removed from published dataset by requiring cells to have at least one count each of *Snap25* and *Slc17a7*. Data was transformed into a Seurat object via *CreateSeuratObject(min.cells = 3, min.features = 200)*, normalized via *NormalizeData()*, and variable genes were identified via *FindVariableFeatures(selection.method = "vst", nfeatures = 2000)*. The Seurat object was then processed according to *ScaleData(), RunPCA(), RunTSNE(), FindNeighbors(), FindClusters(resolution = 0.025), RunUMAP(reduction="pca")* using 50 dimensions, with all other parameters used as default values. This processed Seurat object was then used for downstream analysis. Marker genes were obtained by identifying cluster-enriched genes that obeyed $p_{ADJ} < 0.05$ via *FindMarkers()*, where $p_{ADJ}$ is the adjusted *p*-value from the Benjamini-Hochberg corrected p-value. For pathway analyses, the function DEenrichR-Plot() was used for genes enriched in ovoid cells, with *enrich.database = "GO_Biological_Process_2023"* or "*KEGG_2019_Mouse*", and *max.genes = 300*.

To evaluate whether ovoid cells were present outside of the subiculum, we examined whether adjacent brain regions exhibited ovoid neuron-like transcriptomic profiles. To do this, using a previously published scRNA-seq data[62], a Smart-seq dataset and associated metadata of the cortex and hippocampal formation were downloaded. The data were used to create a Seurat object using *CreateSeuratObject*(min.cells = 3, min.features = 200). Based on the metadata from these cells, cells from the following regions were

extracted and used for analysis: hippocampal region (HIP), para-subiculum, postsubiculum, presubiculum (PAR-POST-PRE), subiculum and prosubiculum (SUB-ProS), retrospenial area (RSP) and restrosplenial area ventral part (RSPv). The cells underwent normalization via *NormalizeData()* and variable genes were identified via *FindVariableFeatures(selection.method = "vst", nfeatures = 2000)*. Then, *ScaleData(), RunPCA(), FindNeighbours(), FindClusters(resolution = 0.15)*, and *RunUMAP()* using 20 dimensions were used, with all other parameters kept at default values. *DimPlot()* employing cluster identities or class/region metadata was used to generate plots. To examine the *Ly6g6e*-expressing cluster in depth, this cluster was subset and analyzed using *FindVariableFeatures(selection.method = "vst", nfeatures = 2000), ScaleData(), RunPCA(), FindNeighbours(), FindClusters(resolution = 0.04), RunUMAP()* using 15 dimensions were used with default parameters unless otherwise noted. *DimPlot()* based on clustering and region metadata was used to generate plots.

## In situ hybridization image acquisition and analysis

To spatially interpret our scRNA-seq results at a single-gene level, we examined marker gene expression via chromogenic in situ hybridization images from the Allen Mouse Brain Atlas[31]. All chromogenic in situ hybridization images are from this Atlas, and include the following gene-experiment pairs from coronal brain sections: *Slc17a7: 75081210, Ly6g6e: 77280582, Cck: 77869074*.

For fluorescent in situ hybridization (FISH), the following probes were purchased from Advanced Cell Diagnostics: *Slc17a7* (Cat No. 317001-T1), *Ly6g6e* (Cat No. 506391-T2), *Cck* (Cat No. 402271-T3), *Gad1* (Cat No. 400951-T4), and *Cre* (Cat No. 312281-T1). mFISH was implemented in a procedurally similar manner to our previously published work[63,64]. First, mature male mice were randomly selected for mFISH and were deeply anesthetized with isoflurane and perfused with phosphate buffered saline (PBS) followed by 4% paraformaldehyde (PFA) in PBS. Brains were dissected and post-fixed in 4% PFA for 24 h, then cryoprotected in a solution of 30% sucrose in PBS for 48 h. Brain sections (20 μm) were made using a cryostat tissue slicer and mounted on coated glass slides. Slides were subsequently stored at −80 °C until use.

For use, the tissue underwent pre-treatment and antigen retrieval per the User Manual for Fixed Frozen Tissue (Advanced Cell Diagnostics). The 4 probes with unique tails (T1–T4) were hybridized to the tissue, amplified, and the tissue counterstained with DAPI. Following probe visualization, the sections were de-coverslipped, fluorophores were cleaved. Some tissue underwent further Nissl staining, wherein sections were incubated in 0.5% PBS-Triton for 10 min, followed by NeuroTrace 60/660 Deep-Red Fluorescent Nissl Stain (ThermoFisher; Cat No: N21483) for 20 min, and then 0.5% PBS-Triton overnight. The sections were then re-coverslipped, re-stained with DAPI and imaged.

FISH images were acquired with a 63× objective on a SP8 Leica white light laser confocal microscope (Leica Microsystems). Z-stacks were acquired with a step size of 0.45 μm for each imaging round. Final composite images are pseudocolored maximum intensity projections, with channels opaquely overlaid upon one another ordered from highest to lowest expression. Presented images include brightness adjustments applied to individual channels uniformly across the entire image, and a linear smoothing filter utilized on probes in the 750 nm channel (*Gad1*-T4) to accommodate noise introduced via increased gain and laser power required in this channel during imaging.

Data was analyzed using our previously published analysis pipeline[63–66], along with extensions to accommodate Nissl- and DAPI-based phenotyping (see mFISH analysis in Supplementary Methods). To register Nissl-stained sections, maximum intensity projections of the DAPI signal from the probe image and Nissl image was used to rigidly register probe signals, followed by nonlinear elastic registration via bUnwarpJ[33,34,56] to accommodate any nonlinear tissue warping due

to de-coverslipping. For automated segmentation of mFISH images, cells were segmented by DAPI by running a Gaussian blur to smooth signal, binarizing the resulting image, and performing segmentation. DAPI-segmented nuclei were dilated by 3 μm to include the surrounding cytosol. Cell bodies used for Nissl-based analysis were manually segmented, such that Nissl-labeled cell bodies that fully encapsulated DAPI-labeled nuclei were traced by hand. This segmentation was restricted to *Slc17a7*-expressing cells to ensure only excitatory neurons were retained for analysis. These manually segmented cells had their height and width with respect to the alveus manually measured. In all cases, the signal from each probe was then binarized by thresholding at the last 0.2–1% of the histogram tail, and then the number of pixels within regions of interest selected from segmentation was summed and normalized to the pixel area of the cell and multiplied by 100. This in effect corresponded to percent area covered (PAC) of the optical space of a cell.

In total, four mature male mice with sections from the anterior dorsal subiculum were used, with a total of 9 sections undergoing FISH (2 with Nissl, and 7 with DAPI only). To facilitate analysis of excitatory neurons specifically, a threshold of 0.3 PAC of *Slc17a7* and 0 PAC of *Gad1* was required for each cell to be included in analysis. For DAPI-segmented cells, this resulted in 142,209 total cells, and 73,861 putative excitatory neurons. For Nissl-segmented cells, this resulted in 301 total putative excitatory neurons. Phenotype indices were calculated via: $(Ly6g6e\text{-}Cck)/(Ly6g6e+Cck)$, with a value of 0 was used to partition *Ly6g6e*-expressing cells and *Cck*-expressing cells. Nissl-stained cell bodies were measured for cell height and width oriented with respect to a horizontal alveus, alveus aspect ratio (cell height/cell width when oriented with respect to a horizontal alveus), area, perimeter, circularity ($4\pi \times \text{area}/(\text{perimeter}^2)$), minimum Feret diameter, maximum Feret diameter, and Feret ratio (minimum Feret diameter/maximum Feret diameter). Principal component analysis was performed on all of the above features using centered and scaled data.

## Generation of Ly6g6e-IRES-Cre mouse line

The Ly6g6e-IRES-Cre targeting construct was produced using recombineering techniques. A 5149 bp genomic DNA fragment containing exon1-3 of the *Ly6g6e* gene was retrieved from BAC clone RP24-271B10. An IRES-Cre-frt-PGKNeoR-frt was inserted immediately after the TAG stop codon. The homologous arms of the construct were 1951 bp and 3190 bp respectively. To facilitate ES cell targeting, Crispr/cas9 system was used. The gRNA sequence is GTTGACAGCCGTGGACGCATGGG and was in vitro transcribed using MEGA shortscript T7 kit (Life Tech Corp AM1354).

The targeting vector, cas9 protein (TrueCut Cas9 Protein V2), and gRNA (concentrations of 10 μg, 3.75 μg and 1 μg respectively in total volume of 100 μL) were co-electroporated into 1 million of G1 ES cells derived from F1 hybrid blastocyst of 129S6 × C57BL/6J. A total of 48 G418 resistant ES colonies were isolated and screened by nested PCR using primers outside the construct paired with primers inside the IRES-Cre-frt NeoR frt cassette. Thirty six of the 48 clones were identified with both arms positive. Three of these ES clones were used for generation of chimeric mice. Chimeric mice were generated by aggregating the ES cells with 8-cell embryos of CD-1 strain. Chimeras were bred with R26FLP (Jax stock 003946, backcrossed to C57Bl/6J for 13 generations to remove the frt-neo-frt cassette.

## Anatomical mapping of bulk ovoid neuron projections

Mature Ly6g6e-IRES-Cre mice of either sex underwent stereotaxic surgery for viral tracing of subiculum cells. During surgery, mice were placed under isoflurane anesthesia maintained at -2.0% and received a local injection of Bupivacaine (2 mg/kg) at the site of incision. To deliver virus, pulled pipettes were backfilled with mineral oil and then loaded with virus. Virus was injected at a rate of 50 nL/min, and the pipette remained at the injection site for 3 min post-injection. All

dorsoventral coordinates were measured from pial surface. Mice received subcutaneous injections of Metacam (5 mg/kg) and 1 cc saline on the day of surgery and for 3 days after.

For anterograde tracing from tissue slices, mice received injections of AAV-CAG-FLEX-EGFP into the subiculum. Two animals were injected unilaterally in dorsal subiculum (−3.6 anteroposterior from bregma, −2.5 mediolateral from bregma, −2.0 dorsoventral; 100 nL), and a third at two depths (−1.5/−2.5 dorsoventral; 80 nL per depth). Another animal received unilateral injections at two sites and two depths per site (−3.3/−3.8 anteroposterior, −2.25/−2.5 mediolateral, −1.5; −2/−1.5; −2.5 dorsoventral; 80 nL per depth). No topographical differences were observed in downstream labeling as a function of injection coordinates. To sparsely label ovoid neurons for brain-wide light-sheet imaging, a dual-recombinase intersectional strategy was used with mice receiving unilateral injections of retrograde rAAV2-retro-EF1a-Flpo into the ATN (−0.7 anteroposterior, −0.7 mediolateral, −3.5 dorsoventral) and AAV8-hSyn-Con/Fon-EYFP into the ipsilateral subiculum at two sites (−3.6/3.9 anteroposterior, −2.65/−3.0 mediolateral, −1.8/−2.0 dorsoventral). For retrograde mapping of ATN-projecting subiculum neurons along with mFISH, two animals were bilaterally injected at two sites per hemisphere (−3.6/−4.0 anteroposterior, ±2.65/±3.0 mediolateral, −1.6/−2.0 dorsoventral; 100 nL/150 nL). To target pyramidal neurons in the subiculum, rAAV2-retro-tdTomato was injected into the nucleus accumbens (+1.3 anterior/posterior, 1.0 mediolateral, −4.0 dorsoventral). No sex differences were observed in anatomical mapping, and mice were pooled across sexes. For comparing ovoid neurons to nucleus accumbens-projecting pyramidal cells, mice were also injected ipsilateral in the nucleus accumbens with rAAV2-retro-tdT, at two different depths with 200 nL/depth (2.0 anteroposterior, 1.0 mediolateral, −5.0; −3.8 dorsoventral). Tissue was taken from these animals and used for virus detection, and subsequent mFISH per published protocols[66].

## Tissue clearing, imaging, and analysis for light-sheet microscopy

SHIELD (Stabilization to Harsh conditions via Intramolecular Epoxide Linkages to prevent Degradation) tissue processing and clearing method was used to render mouse brains transparent[67]. Ready-to-use commercial SHIELD preservation and passive clearing reagents were acquired from LifeCanvas technologies. To ensure tissue preservation before delipidation, resins were embedded in PFA-fixed brains by incubating them with 20 ml of SHIELD OFF solution (containing SHIELD buffer solution, SHIELD Epoxy solution, and distilled water prepared in 1:2:1 ratio respectively) for 4 days at 4 °C with shaking. SHIELD OFF solution was then replaced with SHIELD ON buffer for 24 h at 37 °C with shaking. For clearing, tissue was incubated in 20 ml SDS-based passive clearing buffer at 37 °C for ~3–4 weeks. Samples were checked for extent of clearing of tissue (fading of opaque center) using a sketching light box until they were completely translucent (not transparent). To wash excessive SDS carried over from the clearing buffer, samples were washed with PBST (1% Triton X-100 in 1X PBS) at 37 °C overnight with shaking and stored in 1X PBS + 0.02% sodium azide at 4 °C until ready for imaging.

For optical clearing prior to imaging, the refractive index of the tissue was matched to 1.46 by dipping it in 20 ml of 100% EasyIndex solution (LifeCanvas Technologies) overnight at 37 °C with gentle shaking. The optically transparent tissue was then imaged using the Mesoscale Imaging System (Translucence BioSystems) with a Zeiss Light Sheet Z.1 Microscope System (Z.1 LSM; Carl Zeiss Microscopy, Germany). Cleared whole brain was imaged with cerebellum glued to the sample holder and olfactory bulb hanging towards the base of the chamber filled with Easyindex. Volumetric tiles (spanning entire tissue) were acquired with 488 nm laser (6% power) with both light sheets (online fusion) using the 5X/0.16NA objective (zoom: 0.36; XYZ pixel size: $1.31 \times 1.31 \times 5.03\,\mu m$). The acquired tiles were stitched (with ~100 μm overlap between tiles) with the tile sorter feature of the Arivis

Vision 4D software (Carl Zeiss Microscopy, Germany). Stitched files were 3D rendered and exported as movies with Arivis (Carl Zeiss Microscopy, Germany).

## Analysis of single-cell axonal projections and dendritic morphologies

All single-cell projection analyses used publicly available data from the HHMI Janelia MouseLight database (http://mouselight.janelia.org)[37]. Neurons with a cell body in the subiculum were selected for further analysis, via the Neuron Browser and query selection of a soma in the subiculum. Properties of individual cells were loaded into FIJI[68] according to MouseLight IDs via the Reconstruction Viewer Plugin. For each neuron, analysis of axonal length was conducted with the deepest ontology = 6, and only gray matter structures were included in analysis. For clustering and dimensionality reduction via UMAP, neurons were normalized to have unit length. UMAP dimensionality reduction was performed via the *umap* R package with 20 nearest neighbors, and all other parameters at default values. Hierarchical clustering was performed with Euclidean distances between cells and Ward's distance, with two clusters selected for cutting the tree. Lengths depicted in figure reflect gray matter regions with a minimum mean of 1 mm, to prevent lots of rare/minor projections from being visualized.

All single-cell dendritic morphology analysis used publicly available data from the HHMI Janelia MouseLight database (http://mouselight.janelia.org)[37]. Neurons with a cell body in the subiculum were selected for further analysis, via the Neuron Browser and query selection of a soma in the subiculum. From all cells that had dendritic reconstructions (n = 71), ovoid cells (n = 23) were obtained by examining cells that projected to the anterior thalamic nuclei, whereas all other cells were assigned as pyramidal cells (n = 48). For all cells, dendritic properties (including number of branches and total branch length) were obtained directly from the MouseLight database and analyzed.

## Slice preparations for patch-clamp electrophysiology

Ly6g6e-Cre × Ai14[69] animals were used optically target tdTomato-labeled deep subiculum neurons for patch-clamp recording, and unlabeled patching of non-double-transgenic littermates was also used. In all cases, horizontal slices were prepared from 4-to-8-week-old mice of both sexes. Mice that were up to 5 weeks of age mice were deeply anesthetized, decapitated, and brains were rapidly removed. Mice 6 weeks of age or older additionally underwent transcardial perfusion using ice-cold sucrose-based artificial cerebrospinal fluid (aCSF) containing (in mM) NaCl (85), NaH$_2$PO$_4$ (1), CaCl$_2$ (0.5), KCl (2.5), MgCl$_2$ (7), NaHCO$_3$ (26), sucrose (50), and glucose (10) prior to decapitation. Horizontal slices (300 μm) were cut using a vibratome (Leica VT1200S) in ice-cold sucrose-based aCSF. Immediately after slicing, slices were transferred to a submerged holding chamber and stored in sucrose-based aCSF at 32 °C for 30 min before transfer to room temperature aCSF until recording. Composition of aCSF was as following (in mM): NaCl (129), NaH$_2$PO$_4$ (1.25), CaCl$_2$ (1.6), KCl (3.0), MgSO$_4$ (1.8), NaHCO$_3$ (21), Glucose (10). Slices were transferred individually to a submerged recording chamber and continuously perfused with aCSF at 10–13 mL/min at 30 °C.

## Patch-clamp recording and analysis

Whole-cell patch-clamp recordings of neurons in the subiculum were performed with borosilicate pipettes (1.5 mm outer diameter, Science Products) pulled with a horizontal puller (P-97, Sutter instruments, Novato, CA, USA) (4–6 mΩ). Pipettes were filled with internal solution containing (in mM): K-gluconate (115), KCl (20), phosphocreatine (10), HEPES (10), MgATP (2), NaATP (0.3), EGTA (0.2) and 0.1% biocytin (pH adjusted to 7.4 using KOH, 285 mOsm). In all experiments, access resistance did not exceed 25 MΩ. No series resistance compensation was used. Signals were low-pass filtered at 2 kHz and sampled at 10 kHz

by a Digidata 1550b interface and processed by PClamp11 software (Molecular Devices, Sunnyvale, CA, USA). Ovoid cells were selected based on anatomical location close to alveus, as well as tdTomato expression for Ly6g6e-Cre$^{Tg/+}$/Ai14$^{Tg/+}$ mice. Pyramidal neurons were selected by their location distal to alveus and, in the case of Ly6g6e-Cre$^{Tg/+}$/Ai14$^{Tg/+}$ mice lack of tdTomato expression.

For assessment of intrinsic properties and firing patterns, baseline current was injected to keep cells near a -70 mV baseline potential, and step currents were injected. Input resistance was calculated using the steady-state response to a small negative current injection (−40 to −50 pA). Sag ratio was calculated as the difference between the peak hyperpolarization and steady-state voltage following a −100 pA current injection, normalized by the sag peak, and thus a larger ratio indicates a larger sag. After recording, slices were immediately fixed in 4% PFA overnight for biocytin staining. After fixation, slices were washed in PBS, incubated with Streptavidin secondary antibody (1:500, Streptavidin, Alexa 488 Conjugate, Invitrogen, S32354) at room temperature for 6 h, or 4 °C overnight, and coverslipped using DABCO + PVA (Sigma, Burlington, MA, USA). Biocytin-labeled cells were imaged using a 20× or 63× objective on a Zeiss LSM 880 confocal microscope (Zeiss, Oberkochen, Germany). Data were analyzed using custom R scripts. No sex differences were observed in electrophysiological properties, and mice were pooled across sexes.

## Computational modeling
For computational modeling, morphologically reconstructed subiculum pyramidal and ovoid neurons from the MouseLight dataset[37] were used. Simulations were performed using the NEURON simulation environment (Hines and Carnevale[70]). Model biophysics were based off of previous models of hippocampal pyramidal neurons[71,72], and included two A-type potassium channels (K$_A$) (with distance-dependent channel densities, as described below), a delayed-rectifier potassium channel (K$_{DR}$, uniform density 0.04 S/cm$^2$), a voltage-gated sodium channel (Na$_V$, somatic density 0.09 S/cm$^2$ and elsewhere 0.027 S/cm$^2$), a T-type calcium channel (Ca$_T$, uniform density 0.03125 mS/cm$^2$), and a leak channel (density 0.003125 mS/cm$^2$ within 100 μm of soma, 0.00625 mS/cm$^2$ elsewhere, with reversal potential between −35 mV and −46 adjusted on a per-cell basis to give similar resting member potentials). Consistent with previous models[68], K$_A$ conductances were captured by differential proximal and distal K$_A$ channels (distal K$_A$ conductance: $0.00495 \times (1 + xdist \times 0.01)$ S/cm$^2$, where xdist is the distance in microns from the soma; proximal K$_A$ conductance: 0.00495 S/cm$^2$ within 50 μm of soma; note radial obliques had 0.0198 S/cm$^2$ proximal K$_A$ and distal K$_A$). To generate spike trains, a current step was applied to the soma (500 ms of 150 pA current), with somatic voltages illustrated. To calculate input resistance values, a current step of 5 pA was applied to the soma under the same parameters.

## Surgeries for calcium imaging
Miniscope surgeries involving calcium imaging were performed in two parts. Surgical protocols were generally followed as in anatomical mapping of ovoid neuron projections, except were unilaterally injected with 500 nL of AAV1-Syn-GCaMP6f-WPRE-SV40 or AAV1-Syn-FLEX-GCaMP6f-WPRE-SV40 at 50 nL/min in the dorsal subiculum (+2.5 mediolateral from bregma, −3.7 mm anteroposterior from bregma, and 1.7 mm dorsoventral from the skull surface). Two weeks later, mice underwent a GRIN lens implantation surgery. The skull was scored using a 25-gauge needle, a craniotomy ~2 mm in diameter was performed and the cortical tissue above the subiculum was aspirated using a 27-gauge blunted needle until the corpus callosum was revealed. A GRIN lens (1.8 mm in diameter, 4 mm in length, 0.25 pitch, Edmund optics) was stereotaxically lowered into the targeted implant site. Before imaging, a metal baseplate was fixed to the headcap using dental cement. Once dried, Kwik-Sil was used to cover the GRIN lens to protect it from debris and scratching.

After this, mice were habituated to the handler for 2–3 days and then to wearing the miniscope for 5 days. Habituation occurred for a minimum of 10 min each session to acclimatize animals to the weight of the scope. A wireless miniscope was used for all imaging during behavior[73], powered by a 45 mAh lithium-polymer battery that was affixed to the top of the microscope. For calcium imaging during behavior, per standard miniscope protocols, the miniscope was secured to the baseplate, and a 10 s video was initially recorded before starting the assay to ensure the cellular field of view was unobstructed. After this, the miniscope was powered on and the mouse placed into the center of the arena. After 5 min, the mouse was removed from the arena and miniscope detached. Raw imaging data were saved to a microSD card and transferred to a computer after the session has ended.

## Analysis of calcium imaging data
For analysis of calcium imaging data, the Minian pipeline was used[74]. Using this pipeline, background, and sensor noise were removed, motion correction was performed using a template-matching algorithm, and a seed-based procedure was used to identify spatial and temporal footprints of neurons. Appropriate filter sizes for denoising, cell size during background removal, minimum fluorescence thresholds, and peak-to-noise ratios were determined on a per session basis using the recommendations and visual inspection built into the Minian pipeline.

For analysis using the Minian pipeline, we provide more rigorous detail on the paramters used for these data. All raw data was denoised using a median filter (typical kernel size range: 5–7 pixels) for each recording. A morphological tophat operation was applied to remove background where fluorescence was smaller than largest expected cell (typical range: 10–15 pixels). A max projection was calculated, and local maxima were identified with a rolling window (typical radius range: 10–15 pixels). Peak-noise ratio was then determined via the aid of visualization tools in the Minian pipeline, wherein a low-pass filter (putative signal) and high-pass filter (putative noise) were applied to a subset of seeds, and a noise frequency was chosen that best separated the two signals (typical range: 0.06–0.1). Data then underwent a Kolmogorov-Smirnov test to remove non-bimodal noise-like activity. Data from this pipeline then underwent constrained nonnegative matrix factorization (CNMF), which refined the spatial footprints of cells and denoised the temporal traces of each cell, ultimately outputting calcium traces (variable "C" from Minian), which formed the basis for analysis using in-house R scripts. Parameters selected at the CNMF stage included sparseness penalty (typical range: 0.1–1.0 for spatial updates, 3.0–10 for temporal updates) and noise frequency (typical range: 0.06–0.1), determined on a per-recording basis using Minian visualization tools.

In order to compare ΔF/F and CNMF-produced traces ("C"), ΔF/F was calculated according to ΔF/F = (C−F)/F, with baseline fluorescence F computed in two ways for different analyses. First, for assessing overall cellular activity globally, baseline C values were manually identified on a per-cell basis across the duration of the recording, and used as a global scaling factor to normalize overall activity. To calculate a duration-parameter-free measurement of cellular responses of objects, in the 20-s window following novel object encounters, trial-by-trial peak ΔZ values were obtained on a per-cell basis for both ovoid and pyramidal neurons. Each peak in a given trial was transformed into a percentile relative to a null distribution for each cell, with the null distribution generated by calculating ΔZ peak values across all times. All trial-by-trial percentile values were pooled across ovoid neurons and pyramidal neurons, and compared via Mann-Whitney U test. To determine place tuning, spatial mutual information (MI) was calculated from a binarized deconvolved spiking signal ("S" output from Minian). MI was computed as follows, with X denoting the binned spatial position of the mouse, and K denoting the value of the binarized

spiking signal: $I_{pos} = \sum_{i=1}^{N} \sum_{k \geq 0} P_{x_{i,k}} \cdot \log\left(\frac{P_{x_{i^k k}}}{P_{x_i} \cdot P_k}\right)$. To identify spatial tuning from MI, the MI was calculated between X and K times series using 500 randomly shuffled offsets between time series. A cell was then considered a place cell if its MI score fell above 95% of the shuffled MI values. Speed tuning was assayed by correlating calcium activity with instantaneous speed of the animal. Null distributions for speed tuning were obtained by shifting calcium activity by 10 s relative to instantaneous speed, and recomputing correlations. Cells were determined to be significantly tuned to speed if their speed tuning scores fell outside central 95% of the shuffled speed scores. For computing mobility tuning, a threshold velocity of 1 cm/s was used to assign mobility vs. immobility across a recording, and mean cellular activity was computed for mobility vs. immobility on a per-cell basis. Changes in activity in mobility was then computed according to the difference between mean activity in mobility and immobility, normalized by the mean activity in immobility. Null distributions for mobility tuning were obtained by shifting calcium activity by 10 s relative to instantaneous mobility and changes in activity. Cells were determined to be significantly tuned to changes in mobility if their tuning scores fell outside central 95% of the shuffled speed scores.

Behavioral videos were manually analyzed to parse interaction times. To correlate behavioral interactions with cellular activity, Z-scored (i.e., mean-subtracted and standard-deviation normalized) calcium traces were calculated, averaged in 2 s bins, and averaged across events. To assess changes in activity (ΔZ) driven by object interactions, cellular activity prior to object interactions was subtracted, and cells were assigned a "responder" classification if they surpassed 0.25 ΔZ for a minimum of 10 s. For assessing changes driven by local object interactions, baseline F was taken to be the mean C activity in the 10-s window prior to object activity, with the changes in activity measured as the average change in activity from this baseline in the 20-s window after interaction. For details on the calculation of duration-free measurement of cellular responses, place cell tuning, speed tuning, and mobility tuning, see Supplementary Methods. No sex differences were observed in functional or behavioral properties, and mice were pooled across sexes.

### Novel object recognition and novel object location behaviors

The novel object recognition assay involved training (encoding) and test (retrieval) sessions. In the training session, a mouse was placed in a circular arena (60 cm in diameter) with two, novel objects and allowed to explore for 5 min. After a variable delay (24 h to 100 days later), a test session occurred where the mouse was reintroduced to this object from the training session (familiar object) and a new object (novel object) and allowed to explore for 5 min. Time spent interacting each object was totaled and calculated into a discrimination index (DI), defined according to DI = (T_novel − T_familiar)/(T_novel + T_familiar). Object identities and spatial locations were counterbalanced across animals, with objects and the arena cleaned with 70% ethanol between all sessions.

The novel object location assay was conducted in the same arena and with the same training day protocol as the novel object recognition assay. On the test day, 24 h after training, the mouse was reintroduced to both objects from the training session, but one object was moved to a new location. DI was calculated as above, with novel and familiar location replacing the novel and familiar objects.

### Optogenetic surgeries and behavior

Procedurally, surgeries involving viral injections for optogenetic manipulation proceeded analogously to miniscope viral surgeries. Mice were bilaterally injected with AAV-Ef1a-DIO-ChETA-EYFP (ovoid excitation), AAV-CAG-DIO-ArchT (ovoid inhibition), AAV-CamKII-ArchT-GFP (pyramidal inhibition), AAV-hSyn-ChETA-eYFP (pyramidal excitation), or AAV-CAG-GFP (controls) at the dorsal subiculum (+2.5 mediolateral from bregma, −3.7 mm anteroposterior from

bregma, and 1.7 mm dorsoventral from the skull surface) at 50 nL/min. After virus was allowed to diffuse for 10 min, optic fibers were stereotaxically implanted immediately above the injection site (+2.5 mediolateral from bregma, −3.7 mm anteroposterior from bregma, and 1.6 mm dorsoventral from the skull surface). The skull was scored using a 25-gauge needle and the optic fibers were secured with cyanoacrylate glue and dental cement. Optical fibers were pre-made to reach the dorsal subiculum (1.6 mm dorsoventral from the skull surface), where the bottom of the ferrule would be secured flesh with the skull. Fibers were cut, polished, and inserted into ferrules using epoxy resin. Dental cement used to secure the ferrules was darkened to minimize light leakage. Mice were habituated to the handler for 2–3 days and then to wearing the optogenetic cables for 5 days for a minimum of 10 min each.

Novel object recognition and novel object location behavior paradigms proceeded analogously to those used in calcium imaging. For optogenetic manipulation during the behavioral assay, a sleeve connecting the optogenetic cables to the laser was attached over the ferrules. Once attached, the mouse was introduced into middle of the arena and allowed to explore for 5 min. In inhibition experiments, mice received 20 s pulses of 589 nm light at 15 mW, followed by 20 s of no light, throughout the 5-min session (with the exception of one animal, which received continuous light throughout the duration of the experiment). In stimulation experiments, mice received 10 s pulses of 473 nm light at 1 mW, followed by 20 s of no light, throughout the 5-min session. Behavior in optogenetic-manipulated animals was compared to control animals that had optic fibers implanted. Control animals consisted of animals with no AAV injections (n = 7), as well as animals that had AAV injections of GFP fluorophore only (i.e., lacking opsins; n = 7). Both groups of control animals were handled analogously to opsin-injected animals, with no significant differences present between groups in either encoding or retrieval.

### Quantification and statistical conventions

Unless otherwise noted, the following conventions were used in this study. Box-and-whisker plots show distribution of gene expression within cell populations according to the following conventions: midline denotes median, boxes denote first and third quartiles, whiskers denote remaining data points up to at most 1.5 × interquartile range, with outlier values beyond whiskers not shown. Summary statistics indicate mean ± SEM. Unless otherwise specified, pairwise comparison p values are computed on within-animal-averaged data (typically Mann-Whitney U test, with t-tests used if exceeding 100 observations per group). Statistical significance for adjusted p values is denoted as: ns: $p \geq 0.05$; *: $p < 0.05$, **: $p < 0.01$, ***: $p < 0.001$.

### Reporting summary

Further information on research design is available in the Nature Portfolio Reporting Summary linked to this article.

## Data availability

The data generated in this study are provided in the Supplementary Information/Source Data file. Source data are provided with this paper.

## Code availability

Code from this manuscript are provided publicly through Figshare (https://figshare.com/projects/Atypical_hippocampal_excitatory_neurons_express_and_govern_object_memory/223434).

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

## Acknowledgements

M.S.C. is supported by the University of British Columbia (Department of Cellular and Physiological Sciences, Djavad Mowafaghian Centre for Brain Health, and the Faculty of Medicine Research Office), the Natural Sciences and Engineering Research Council of Canada (RGPIN-2019-04507), the Canadian Institutes of Health Research (PJT-419798 and PJT-183950), and the Canadian Foundation for Innovation (John R. Evans Leaders Fund 38369). A.I.K., S.R.E., R.E.C., K.E.S., M.K., and B.N.B. are all supported by a Canada Graduate Scholarship—Master's from the Canadian Institutes of Health Research. A.I.K., K.E.S., and B.N.B. are all supported by a Canada Postgraduate Scholarship—Doctoral from the Natural Sciences and Engineering Research Council. S.R.E. is supported by the Cordula and Gunter Paetzold Fellowship from the University of British Columbia. B.N.B. is supported by the Dorothy May Ladner Memorial Fellowship. L.K. is supported by the Deutsche Forschungsgemeinschaft (DFG, German Research Foundation, Walter Benjamin fellowship, project 444112617). M.S.C. and C.G. are supported by funding through the Howard Hughes Medical Institute. This work was supported by resources made available through the NeuroImaging and NeuroComputation Core at the Djavad Mowafaghian Centre for Brain Health (RRID: SCR_019086). We thank members of the Cembrowski lab for helpful discussions, and Jeffrey LeDue for insight and guidance in image acquisition.

## Author contributions

M.S.C. provided supervision, conceptualized the experiments, oversaw project administration, and acquired funding. A.I.K., D.N.M., S.R.E., C.N.M., R.E.C., K.E.S., L.K., M.K., B.N.B., M.Y.Z., M.W.E., S.C.W., A.T., E.K., W.D., M.A., C.G. set up methodology and/or conducted experiments. A.I.K., D.N.M., S.R.E., R.E.C., K.E.S., M.K., M.W.E., J.T., and M.S.C. conducted data analyses and visualization. A.I.K., D.N.M., S.R.E., K.E.S., and M.S.C. wrote and edited the manuscript with additional reviewing and editing by all other authors.

## Competing interests

The authors declare no competing interests.
