## [Transparent Peer Review file · Nature Communications]

Atypical hippocampal excitatory neurons express and govern object memory

Corresponding Author: Professor Mark Cembrowski

Version 0:

Reviewer comments:

Reviewer #1

(Remarks to the Author)

In this study, Kinman et al provided evidence of a non-typical subtype of subiculum neurons structurally and functionally different from the classical subiculum pyramidal neurons. These neurons had “ovoid” shaped cell bodies. The ovoid neuron activity is attributed to a selective non-spatial form of novelty. Whereas it is less responsive to familiar object exploration. The authors generated a transgenic Ly6g6e-IRES-Cre mouse line for genetic labeling, calcium imaging, and further optogenetic manipulation of these ovoid neurons. The authors demonstrated that optogenetic inhibition of ovoid neurons marks memory encoding deficits while silencing subiculum pyramidal neurons evoked deficits in spatial location memory. The experiments are well thought out, and the findings are novel. However, a few comments need to be addressed:

1. How unique are these ovoid cells in the mouse brain? Do they appear in other brain regions that are closely connected with the subiculum such as CA1 and retrosplenial cortex? Given the proximity of the subiculum with the retrosplenial cortex, does the manipulation of ovoid cells also impact cells in the retrosplenial cortex? Additionally, expression of the AAV across these regions needs to be provided.
2. All the experiments involving the manipulation of ovoid neurons were performed using the Ly6g6e promoter-driven expression of transgenes. Therefore, the expression profile of Ly6g6e across the hippocampal sub-regions both in terms of single-cell transcriptomic analysis as well as in situ analysis needs to be provided. Does this gene express in other hippocampal or cortical cells? Also, why Ly6g6e gene was selected for labeling these “ovoid” neurons amongst 248 enriched genes? The broader role of this gene could be discussed in terms of its molecular function and cellular activity.
3. Silencing ovoid neurons selectively produced a novel object memory deficit, but activation produces familiarity-seeking behavior. The apparent familiarity-seeking behavior needs to be justified while establishing a causal relationship between “ovoid” neurons and non-spatial novelty-seeking. What happens when optogenetic stimulation is stopped during memory retrieval (1 day)? Does it show a reversal to novelty seeking? Inherently, rodents exhibit novelty exploring behavior. Does the “ovoid” neuron stimulation inhibit or overpower the inherent novelty-seeking behavior?
4. In Line 266, the authors mention that “inhibition during retrieval session produced no apparent discrimination impairment (Fig 6D)”. However, Fig 6D shows clear impairment upon inhibiting ovoid cells during the train and test session compared to controls. In the novel object behavioral data presented in Figures 6C and 6D where either ovoid or pyramidal neurons were inhibited optogenetically during training or testing, no significant difference in discrimination was seen between these two groups. Only ovoid neuron inhibition showed significant difference compared to controls. Is this due to small number of sample size used in these behavioral experiments?
5. In the spatial location assay presented in Fig 6R-S, why stimulation of pyramidal neurons did not affect the spatial location memory? Silencing pyramidal cells evoked both encoding and retrieval deficits, but stimulation of pyramidal cells did not show enhanced behavioral performance. As pyramidal cells are important for both encoding and retrieval of spatial location memory, why number of pyramidal responder cells were not altered in extended figure 4F?
6. It is not clear why non-neuronal cells and interneurons were removed while analyzing the single-cell RNA-seq dataset. It would be interesting to investigate the difference in gene expression of the “ovoid” neuron sub-type amongst all other clusters.

7. In the methodology section (lines 853-854), retrograde AAV-EF1a-Flpo and AAV8-hSyn-Con/Fon-EYFP are mentioned but those have no mention in the main text. Whereas rAAV2-retro-tDT for labeling subiculum pyramidal neurons had no mention in the methodology. Is there any part being missed in the manuscript?

Minor comments:

1. The type of statistical test performed in each figure could be mentioned in the figure legend.
2. Were any sex-specific biases observed in the behavior or electrophysiological assays?
3. For broader readership, it should be mentioned why rAAV2-retro-tDT virus was used and why has it been injected into the nucleus accumbens. It is not clear in the text.
3. In some instances it was found that the figures were not explained in detail. For example figure 6R,S, and extended data Figure 4F.

Reviewer #2

(Remarks to the Author)

In Kinman, et. al., the authors investigate a specific, unique neuronal cell type in the subiculum of the hippocampus using a variety of techniques. They reanalyze previously published snRNA-seq from their group, identifying a cell type characterized by the expression of Ly6g6e. They confirm that these neurons are specifically localized to the deep layers of the subiculum and have an ovoid shaped cell body. They then generated a transgenic mouse line to virally label and manipulate Ly6g6e-expressing neurons. They find that ovoid neurons specifically project to thalamic nuclei, have unique electrophysiological properties, and behavioral relevance. Overall the manuscript is well written, uses appropriate techniques and analyses, and offers experimental support for the claims. This neuronal subtype has not been previously described in the hippocampus, and thus these findings are novel and exciting. I have a few comments to improve the manuscript, and find it unfortunate that the authors do not appear to have deposited their analysis code publicly. This should be address prior to publication.

Major:

- Comparison of the UMAP between Figure 1B and 2G is not mathematically justified
- How soma shape (ovoid vs. pyramidal) was characterized for the MouseLight data was not clear to me, and how that thus led to the classification of all ATN-projecting neurons as ovoid was also unclear. Overall I felt that the MouseLight data could be better integrated into the manuscript if the examination of this data was what prompted the authors to further investigate this neuronal cell type.
- Data and code availability: all code used to recreate these analyses should be made publicly available on Github, particularly for analysis of electrophysiology and calcium imaging data.

Minor:

- Manually segmented cell bodies (1L): please clarify how this was done (manual tracing of each individual cell body)?
- There are some discrepancies on the ages of mice used for acute slice preparations, and some missing phrases, likely from repeated rounds of iterative editing, in the Methods section "Slice preparations for patch clamp electrophysiology"
- In the electrophysiological data, was membrane capacitance calculated? This could be of interest given the unique shape and arborization.
- The authors should elaborate on their justification for the choice of the behavioral paradigm used.

Reviewer #3

(Remarks to the Author)

The manuscript by Kinman et al. describes a novel population of excitatory neurons in the subiculum – the 'ovoid' neuron. They first identified a group of neurons showing differential gene expression compared to other subicular excitatory neurons and found that Ly6g6e could be used to define these neurons. They found that Ly6g6e-expressing neurons in the subiculum showed an ovoid shape instead of a pyramidal one and were located in the deepest layer of the subiculum. They further characterized the anatomical and electrophysiological properties of the ovoid neurons and found that these neurons are regular-spiking and have specialized dendritic arbors. Ovoid neurons also primarily projects to ATN. Functionally, the activity of ovoid cells seems to specifically respond to novel objects. Silencing ovoid neurons blocked the encoding of novel object recognition, while activating ovoid neurons evoked familiarity-seeking behavior. Overall, this is a very interesting study describing a novel hippocampal cell type with specialized anatomical, physiological, and functional roles. However, I have some major concerns regarding how well the data support the conclusions, which need to be addressed by the authors in order to publish.

1. It would be helpful to include GABAergic neurons in Figure 1A and 1B to illustrate the separation and differential gene expression patterns between excitatory and inhibitory neurons. More importantly, Figure 1H should provide quantification across different mice to assess the overlap between Ly6g6e and GAD. Currently, the authors showed that the Ly6g6e+ excitatory neurons in Figure 1A are devoid of GAD1 expression, but this does not guarantee that GABAergic neurons do not express Ly6g6e, as Figure 1H is not convincing without quantification. This is important because Ly6g6e was used to define ovoid neurons for the rest of the manuscript.

2. Figure 1K CCK ISH is hard to see.

3. The statistical analyses in Figure 1M and 1O treated individual cells from an unknown number of mice as independent data points. This approach can result in exaggerated p-values and may lead to inaccurate results, especially considering that these neurons share a similar genetic background and should not be treated as independent data points. This issue appears to be universal throughout the entire manuscript. The authors should use each mouse as an independent observation by averaging the cell measurements within each mouse. Alternatively, the authors could use mixed-effects models (Yu et al., 2022, Neuron) to perform statistics, accounting for the data dependence within each animal.
4. As the authors generated this Ly6g6e-IRES-Cre mouse line, which could be a very useful tool for future research, they should provide more verification of this mouse line. For example, 1) show the distribution of Ly6g6e neurons in the subiculum along both the septotemporal and proximodistal axes by crossing with a reporter line and presenting a few coronal sections from the anterior to posterior hippocampus. 2) demonstrate the fidelity of Ly6g6e expression in the Cre+ neurons.
5. Figures 2A-D: It appears that Ly6g6e+ neurons are distributed uniformly across the proximodistal axis of the subiculum (Figure 1F). In the methods, the authors injected AAV into slightly different locations of the subiculum across different mice. The authors should address whether the projections are topographically organized based on the different injection sites. They should also specify how many mice were used for this experiment in the main text or figure legend to ensure that the conclusion is robust and reliable. Additionally, the authors should clarify in the main text whether targeting ovoid neurons at different locations along the proximodistal axis of the subiculum yielded the same results.
6. Extended Data Figure 1H is hard to see.
7. The authors claim that Ly6g6e neurons are smaller than pyramidal neurons. However, this size difference is not obvious in Figure 2B and 3A, comparing with NAc-projecting subicular neurons. Although Figure 3B provides quantification, the statistical methods need to be revised as previously described in point 3.
8. Figure 3E needs to be better contrasted.
9. Figure 3F has the same statistical problem, unless this quantification was based on the data from MouseLight. Then it should be clearly indicated in the legend.
10. Figure 4: please clearly indicate how many mice were used for slice electrophysiology.
11. Figure 5: The criteria for categorizing 'responder' vs. 'non-responder' cells are biased towards long-lasting activity from ovoid cells. The 10-second duration criterion eliminates short-duration responses from pyramidal neurons, resulting in a very low number of responder cells within the pyramidal population. In fact, short-duration activity during the exploration of objects could also contribute to the encoding of objects. The authors should not impose such a long-duration criterion to categorize neurons. Instead, they should vary their criteria and assess whether the observations between ovoid and pyramidal cells hold true regardless of the criteria used. This will ensure that the findings are not biased by arbitrary thresholds.
12. Extended Data Figure 4: Panels G, H, and I, should use individual mice as independent data points for each quantification. The authors did a good job in Figure 5H regarding using mice as independent data points, but I am not sure why they switch the basis for quantification back and forth in different panels.
13. Extended Data Figure 4G: the lack of spatial tuning in both ovoid and pyramidal cells is problematic. Despite the ref cited by the authors, neurons in the subiculum, similar to those in CA1, should primarily tune to spatial positions within an environment (Sharp and Green, 1994, Ledergerber et al., 2021, Sun et al., 2024). These studies consistently report that over 50% of subicular neurons carry significant spatial information. The authors should instead follow Skaggs et al. 1993 and 1996 to use information score to characterize place cells in the subiculum. Additionally, another useful measurement to consider is the firing rate of ovoid versus pyramidal neurons in the open field arena. Ovoid neurons should show a lower firing rate than pyramidal neurons if they specifically encode novel objects.
14. The description of Extended Data Figure 4M-N in the main text is not sufficient for me to understand the goal of the experiment. It seems to me that the mice explored the same set of objects for two days, so on the second day when the mice were in the novel environment, the objects were not novel anymore but there were still significant number of responders within the ovoid cell population?
15. Figure 6C-D: the current interpretation ignored the significant differences between control vs. ovoid in panel D. I think the interpretation should be that ovoid neurons contribute to novel object recognition during both encoding and retrieval stages, but more significant during the encoding and less so during the retrieval.
16. Line 295 - 297: the results from pyramidal neuron stimulation are the same as the control, the author should not imply that stimulating pyramidal neurons evoke novelty seeking.
17. It is interesting to see that activating ovoid neurons evoked biased exploration towards the familiar object. On the other hand, this is counterintuitive to the role of novelty coding of ovoid neurons. The author should further discuss these results and propose some potential interpretations for these interesting but opposing results.

Minor

1. Title: the term 'non-spatial memory' is too broad to refer to 'novel object recognition' or 'novel object encoding', the authors should use more appropriate words to accurately describe the findings based on their experiments and results.
2. Line 51: 'objective environment' is also a vague term that does not accurately describe the role of hippocampal representation.
3. Line 58-59: Works from PE Sharp and DA Nitz should be cited for spatial and trajectory coding in the subiculum.
4. Line 369: does not?

Reviewer #4

(Remarks to the Author)

Version 1:

Reviewer comments:

Reviewer #1

(Remarks to the Author)

The authors have addressed all my prior questions. I have no additional concerns. I congratulate the authors for such an interesting work.

Reviewer #2

(Remarks to the Author)

The authors have addressed my concerns, and I am in support of publishing this manuscript.

Reviewer #3

(Remarks to the Author)

The manuscript is significantly improved, and the authors have addressed most of my concerns. However, they still need to further address or discuss the following two points.

1). Figure 1K CCK ISH is still hard for me to see.

2). Regarding place tuning in the subiculum, I believe the authors may have misunderstood my point. First, it is important to distinguish that significant place tuning is not equal to sharp place tuning. A neuron with a large (not sharp) place field, or even multiple place fields, can still be characterized as a place cell as long as its score passes the shuffle test. The authors' current conclusion is that subiculum neurons (both pyramidal and ovoid) lack sharp place field tuning compared to CA1 place cells. However, the data from this manuscript (5-7% of characterized place cells in the subiculum) support a broader conclusion that subiculum neurons lack any place tuning, whether sharp or broader. This lack of place tuning is inconsistent with findings from previous studies I cited in my previous response.

More specifically, Sharp and Green, 1994 reported that "The majority of cells throughout the transverse extent of the dorsal subiculum showed strong location-related modulation of their firing rates. Eighty-three of the 99 cells showed significant ($p < 0.01$) R values for the spatial coherence measure, with the average R value for all cells being $0.48 (\pm 0.02)$."

In Figure 4E of Ledergerber et al. 2021, just the PHS group itself (regardless of the presence of covariates, any group with letter P here indicates that their activity is significantly spatially tuned) accounts for 53.7% of the total recorded subiculum neurons. Adding the PHS, PH, PS, and P groups together, at least 80% of subicular neurons showed significant place tuning.

In Figure 2I of Sun et al., 2024, roughly 50 – 80% of subicular neurons can be characterized as place cells. The authors also noted that they used the same shuffling approach as in this paper.

Together, all these studies indicate that the majority of subicular neurons show significant place tuning, regardless of the sharpness of their tuning. By contrast, again, the 5-7% of place cells in the subiculum reported in this manuscript suggests minimal place tuning in the subiculum, which is inconsistent with previous studies. I am not suggesting that the authors are wrong, but rather that they should make appropriate conclusions in the main text and discuss potential causes - whether experimental, technical, or analytical - for this discrepancy in place tuning in the subiculum.

Reviewer #4

(Remarks to the Author)
